# DAPE: Data-Adaptive Positional Encoding for Length Extrapolation

**Chuanyang Zheng**[1]*† **Yihang Gao**[2]* **Han Shi**[3] **Minbin Huang**[1] **Jingyao Li**[1]
**Jing Xiong**[4] **Xiaozhe Ren**[3] **Michael Ng**[5] **Xin Jiang**[3] **Zhenguo Li**[3] **Yu Li**[1]
[1]CUHK [2]NUS [3]Noah's Ark Lab [4]HKU [5]HKBU

## Abstract

Positional encoding plays a crucial role in transformers, significantly impacting model performance and length generalization. Prior research has introduced absolute positional encoding (APE) and relative positional encoding (RPE) to distinguish token positions in given sequences. However, both APE and RPE remain fixed after model training regardless of input data, limiting their adaptability and flexibility. Hence, we expect that the desired positional encoding should be data-adaptive and can be dynamically adjusted with the given attention. In this paper, we propose a **D**ata-**A**daptive **P**ositional **E**ncoding (DAPE) method, which dynamically and semantically adjusts based on input context and learned fixed priors. Experimental validation on real-world datasets (Arxiv, Books3, and CHE) demonstrates that DAPE enhances model performances in terms of trained length and length generalization, where the improvements are statistically significant. The model visualization suggests that our model can keep both local and anti-local information. Finally, we successfully train the model on sequence length 128 and achieve better performance at evaluation sequence length 8192, compared with other static positional encoding methods, revealing the benefit of the adaptive positional encoding method.

## 1 Introduction

Transformer-based models have shown state-of-the-art performances in many language processing tasks, including translation [6], question-and-answer [82, 29, 3], and commonsense reasoning [65]. The transformer mainly consists of attention block, feed-forward block, and positional encoding. Recent works [8] have proved that quadratic-cost attention from the softmax is necessary for better performance, especially in long-context processing. The attention block was originally designed by applying softmax to the key-query multiplication, which requires quadratic computational cost. To address such challenges, some efficient transformers were proposed, including sliding window transformers (e.g., Streaming LLMs [77]), linear transformers (e.g., Performer [17]), and sparse transformers (e.g., Reformer and sparse Sinkhorn transformer [66, 25]), etc. However, some negative results exist regarding efficient transformers' performances [80].

It has been noticed recently that well-designed positional encoding significantly improves the model performances, especially in the long-context tasks [33]. While transformer-based models exhibit satisfying performances in tasks of consistent length and distribution, their effectiveness tends to diminish sharply when the input length exceeds the training length, e.g., long document summarization, "needle in a haystack" search, and long text generation. To avoid the expensive computation in training, the training length is usually preferred to be relatively small due to the quadratic cost of

---

*Equal Contribution

†Contact Email: cyzheng21@link.cuhk.edu.hk

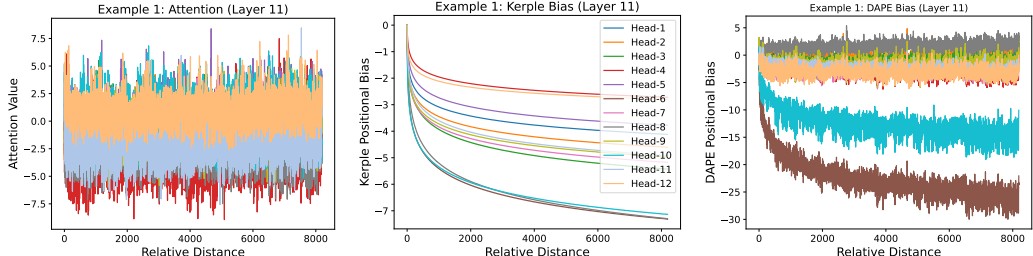

Figure 1: **Visualization of DAPE learned positional biases for the 8192th query position with key positions between 1 and 8192, while the training length is 512**. We notice that DAPE learns both local and anti-local position patterns. The model is trained with Equation 2: (1) The Attention is $\boldsymbol{X}\boldsymbol{W}_Q(\boldsymbol{X}\boldsymbol{W}_K)^\top$; (2) The Kerple bias is $\boldsymbol{B}$; (3) The DAPE (with Kerple) bias is $f(\boldsymbol{X}\boldsymbol{W}_Q(\boldsymbol{X}\boldsymbol{W}_K)^\top, \boldsymbol{B})$. More examples are shown in Appendix I

softmax-based transformers. However, real-world applications often require processing longer input sequences, posing a significant challenge. Therefore, there is a growing interest in evaluating model performance by training on shorter sequences while testing on longer inputs. Standard transformers may not distinguish the ordering of tokens without external assistance. In practice, they depend on positional encoding to incorporate positional information, enabling the model to make meaningful token predictions. Without these encodings, token generation would lack the necessary contextual order, rendering the outputs nonsensical. The RoPE [62] positional encoding method demonstrated a notable performance degradation, failing entirely when the input length is double that of the training length [51, 10, 24]. A common characteristic among these positional encodings is their pre-defined and static nature. Specifically, they are fixed across various tasks and models, which may lead to their inability to adapt to varying input lengths and contexts effectively. To address this issue, recent works have introduced Functional Interpolation for Relative Positional Encoding (FIRE) [41], which utilizes a neural network to learn an implicit mapping from input positions to positional bias. A functional approach to positional encoding that dynamically adjusts positional biases based on semantic information (input context) allows the model to empower adaptability beyond the fixed inductive bias as adopted in previous studies (such as RoPE [62] and Alibi [52]). Although FIRE utilizes MLPs to learn positional embeddings, these embeddings remain fixed across different tasks once the training is completed. Intuitively, the learned static positional encoding (such as Kerple and FIRE) is an average optimal solution across all training samples. Consequently, while they might be generally effective, they are inherently suboptimal for any specific instance. This static nature limits their flexibility and applicability in various real-world scenarios that deviate from the training context. In this paper, we introduce a data-adaptive positional encoding (DAPE) method, inspired by the limitations of static PEs. DAPE dynamically adjusts the PE based on the semantic information (e.g., the current attention value) $a$ and the positional indicator $b$. The proposed PE is represented by MLPs due to their universal approximatability, i.e., MLPs$(a, b)$. We note that DAPE is compatible with all additive relative PEs and offers advantages in terms of interpretability and ease of implementation. The proposed DAPE incorporates both the semantic and the positional information, making the PE adaptive with the input data. The adaptivity allows DAPE to overcome the inflexibility and achieve relatively optimal performance for each individual instance by dynamically adjusting on each specific input data. To the best of our knowledge, this is the first semantically dependent and adaptive positional encoding method introduced in transformer architectures.

The paper is organized as follows. In Section 2, we review some related works on positional encoding methods, including absolute and relative positional encodings as well as the potentially no positional encoding in some transformer models. In Section 3, we introduce the proposed DAPE method with implementation on multi-head attention and analysis on computational costs. We conduct comprehensive experiments on DAPE, validating its effectiveness and performances on various language tasks and datasets, as reported in Section 4. In Section 6, some concluding remarks and potential future works are presented.

## 2   Related Works

**No positional encoding**. Haviv et al. [30] show that decoder-only Transformers with causal attention masks can learn positional information even without any explicit positional encoding. Recently, Kazemnejad et al. [33] proved the effectiveness of no positional encoding (NoPE) [71]. Although

the NoPE can implicitly catch the positional information, it performs poorly compared with some explicit positional encoding methods [41].

**Absolute positional encoding**. Vaswani et al. [69] proposed Absolute positional encoding (APE) to endow transformers with positional information. In particular, in the first layer, a (learnable or fixed sinusoidal) real-valued encoding [69, 35, 42, 70, 47] $e_i \in \mathbb{R}^d$ is assigned to each position $i$, leading to an APE matrix $\boldsymbol{E} = [\boldsymbol{e}_1, \cdots, \boldsymbol{e}_n]^\top$, which will be added to the input sequence. Though simple and straightforward, APE-based Transformers usually generalize poorly to longer sequences [52].

**Relative positional encoding**. Relative Positional Encoding (RPE) is another popular way to encode positional information [58, 56, 52], One popular RPE method in large language models is rotary positional encoding (RoPE) [62, 18, 67]. RoPE rotates the query and key vectors with an angle proportional to their absolute positions before the dot product attention, which results in attention being a function of the relative distance between the tokens, capturing the relative positional information. Press et al. [52] and Kazemnejad et al. [33] found that RoPE-based language models have poor length generalization. To address this, positional interpolation (PI) [11] is proposed to extend the context window. Following the direction, there are LongLora [12], LongRope [24], YaRN [51] and CLEX [10]. Another popular direction is additive positional encoding. For most of these additive RPE methods, the computation of the (pre-softmax) attention logits can be unified using the following formula:

$$\boldsymbol{A}_{\mathrm{RPE}}(\boldsymbol{X}) = \boldsymbol{X}\boldsymbol{W}_Q(\boldsymbol{X}\boldsymbol{W}_K)^\top + \boldsymbol{B},$$

where the bias matrix $\boldsymbol{B} \in \mathbb{R}^{n \times n}$ is induced by the position encoding function $b : \mathbb{N}^2 \to \mathbb{R}$ and the $(i, j)$-th entry of $\boldsymbol{B}$ is defined as $b(i, j)$. Different formulations and parameterizations of $b$ lead to various RPE variants. Several methodologies that facilitate arbitrary sequence lengths include T5's RPE [56], Alibi [52], Kerple [13], Sandwich [14], and FIRE [41]. Currently, additive RPE delivers relatively robust performance in length extrapolation without necessitating additional operations. Alibi constructs the bias matrix $\boldsymbol{B}$ utilizing prior knowledge, resulting in a basis matrix devoid of learnable parameters [52]. Conversely, both Kerple [13] and Sandwich [14] incorporate two learnable parameters to facilitate the learning of a bias matrix while retaining some elements of priors. FIRE suggests adopting a learnable continuous function, such as MLPs, to convert input positions to biases [41]. Observing these developments, it becomes evident that the next generation of bias matrices will likely incorporate adaptivity and flexibility. Based on this insight, we propose our method DAPE, a semantically adaptive method.

## 3 Method

In this section, we formally introduce DAPE (data-adaptive positional encoding), a new relative positional encoding approach that further enhances transformer performance. Compared with previous works on static positional encoding methods, DAPE adopts semantically adaptive positional bias matrices depending on input context. We first demonstrate that most of the popular positional bias matrices are fixed once the training is finished, independent of the input sequences. To address this limitation, we then accordingly develop DAPE that captures the implicit relationships by MLPs and adjusts the bias matrices based on input context. Furthermore, we discuss a variant of DAPE with residual connections and its extensions to multi-head transformers.

### 3.1 Additive Relative Positional Encoding

For most additive RPE methods, the computation of pre-softmax attention logits can be unified under the following formula:

$$\boldsymbol{A}_{\mathrm{RPE}}(\boldsymbol{X}) = \boldsymbol{X}\boldsymbol{W}_Q(\boldsymbol{X}\boldsymbol{W}_K)^\top + \boldsymbol{B}, \tag{1}$$

where the bias matrix $\boldsymbol{B} \in \mathbb{R}^{n \times n}$ is induced by the position encoding function $b : \mathbb{N}^2 \to \mathbb{R}$ and the $(i, j)$-th entry of $\boldsymbol{B}$ is defined as $b(i, j)$. Various formulations and parameterizations of $b$ give rise to different variants of RPE. Examples of additive RPE include: (1) Alibi: $b(i, j) = -r|i - j|$, with the scaler $r > 0$ as a hyper-parameter; (2) Kerple: $b(i, j) = -r_1 log(1 + r_2|i - j|)$ with $r_1$ and $r_2$ are two learnable parameters; (3) FIRE: $b(i, j) = f_\theta \left( \frac{\psi(i-j)}{\psi(\max\{L, i\})} \right)$, where the positional encoding function $f_\theta$ parameterized by $\theta$ is learned from data and $\psi$ is a transformation function aimed at assigning more model capacity to local positions.

We observe from the formulation of those additive RPEs that they remain static once the training process is completed and depend solely on the positions, regardless of the input context. This inflexibility and lack of adaptivity can lead to performance degradation, especially in tasks involving long-context generation and reasoning. Intuitively, the learned static RPEs are optimal on average across all training samples. However, this means they are suboptimal when considering each individual instance, as they cannot adapt to specific tasks. To address these challenges and enhance model performance, it is essential to adopt an alternative approach using a semantically adaptive RPE that depends on input context.

## 3.2 Data-Adaptive Positional Encoding

For simplicity, we first consider the single-head case and the extension to the multi-head transformer will be discussed subsequently. The design of data-adaptive positional encodings in natural language tasks is motivated by the need to capture the intricate relationships between tokens. Arora et al. [5] reveals that associate recall accounts for most of the perplexity difference between transformer, RNN-based, and convolution models. For example, we consider a consistent pairing that "Hakuna" is always followed by "Matata" in a long paragraph. This pattern suggests a reduced reliance on positional information in favor of enhancing token embedding similarity, thus allowing for 'Hakuna' to be effectively linked with a preceding 'Matata'. Similarly, in tasks involving long-context understanding and search, semantic similarity should be prioritized in the attention mechanism rather than being overshadowed by positional encodings, which can be less relevant over long distances. Consequently, the transformer should preserve information without being influenced overly by positional distance. Instead, a satisfactory PE should integrate both semantic and positional information. Therefore, a semantically dependent positional encoding approach is preferable and expected to enhance model performances. Here, we use the attention $\boldsymbol{X}\boldsymbol{W}_Q(\boldsymbol{X}\boldsymbol{W}_K)^\top$ to represent the semantic information and positional bias matrices $\boldsymbol{B}$ (e.g., Alibi and FIRE) to capture positional details. Then the context-adaptive PE is described by $f(\boldsymbol{X}\boldsymbol{W}_Q(\boldsymbol{X}\boldsymbol{W}_K)^\top, \boldsymbol{B})$, where $f(\cdot)$ is an implicit function that integrates both semantic and positional data into the desired positional encodings. Thus, the pre-softmax attention logit incorporated with DAPE is formulated as

$$\boldsymbol{A}_{\mathrm{DAPE}}(\boldsymbol{X}) = \boldsymbol{X}\boldsymbol{W}_Q(\boldsymbol{X}\boldsymbol{W}_K)^\top + f(\boldsymbol{X}\boldsymbol{W}_Q(\boldsymbol{X}\boldsymbol{W}_K)^\top, \boldsymbol{B}). \tag{2}$$

Here, $f : \mathbb{R}^{T \times T} \times \mathbb{R}^{T \times T} \to \mathbb{R}^{T \times T}$ is an element-wise function. In practice, we utilize a two-layer *LeakyReLU* neural network to parameterize $f(\cdot)$ due to its universal approximability [36]. All parameters are learned directly from the data during the training process. This architecture allows $f(\cdot)$ to dynamically adjust positional embeddings based on the input context, ensuring that the encoding method is both adaptive and dependent on the input data.

Different from FIRE, which also models the implicit positional encoding by MLPs, our approach additionally integrates semantic information. This integration enables the adaptivity, flexibility, and context-dependency of the positional encodings. Significantly, our method is compatible with most additive RPE techniques, as these commonly involve positional bias matrices $\boldsymbol{B}$ that inherently contain positional relations. Unlike previous RPEs, which rely solely on absolute positional differences, our DAPE method, can be seen as utilizing multi-level positional bias matrices. Here, the bias matrices dynamically adjust based on the input context, offering a more reasonable and responsive encoding mechanism.

**Expressiveness**. Due to the universal approximability of (*LeakyReLU*) neural networks [36], $f(\cdot)$ is capable of capturing complex relationships between the desired positional encoding and both semantic and positional information. Regardless of the semantic component, when the relative position $i - j$ is used as input, DAPE can realize classical additive RPEs (e.g., Alibi and Kerple), according to [41]. This demonstrates the versatility of DAPE in accommodating traditional encoding schemes while also offering enhanced capabilities. There exists a fundamental trade-off between the expressiveness and computational costs. Wider hidden layers lead to higher expressiveness but also contribute to more computational costs. In practice, we find that two-layer neural networks with 32 hidden units per layer provide sufficient expressiveness to deliver satisfactory performance, balancing complexity and efficiency effectively.

**Discussion.** We can also interpret the proposed method from an alternative perspective. In the standard transformer architecture, the pre-softmax attention typically involves the key-query similarity and the positional encoding by either addition (in the form of $a + b$, e.g., Alibi and Kerple) or multiplication (in the form of $a * b$, e.g., RoPE). Here, we propose a unified approach by replacing them with

learnable MLPs, i.e., MLP$(a, b)$. This configuration allows the model to learn the desired relationship between the pre-softmax attention, the key-query similarity and the positional encoding. It can also be regarded as a new transformer architecture that empower the transformer with additional MLPs on pre-softmax attentions.

**A variant of DAPE with residual connections**. It is well-known that deep neural networks may suffer from gradient vanishing. To further enhance the practical performances, we introduce the residual connection for positional information. Consequently, Equation 2 is modified as follows:

$$\boldsymbol{A}_{\text{DAPE}}(\boldsymbol{X}) = \boldsymbol{X}\boldsymbol{W}_Q(\boldsymbol{X}\boldsymbol{W}_K)^\top + \boldsymbol{B} + f(\boldsymbol{X}\boldsymbol{W}_Q(\boldsymbol{X}\boldsymbol{W}_K)^\top, \boldsymbol{B}). \tag{3}$$

In this reformulation, $f(\cdot)$ acts as an adaptive correction term to the traditionally fixed RPE, dynamically adjusting the positional bias matrices $\boldsymbol{B}$ based on both semantic and positional inputs. In Section 4, we empirically explore the impact of residual connections in DAPE. Our observations reveal that for well-behaved bias matrices $\boldsymbol{B}$, the DAPE model with residual connections, as specified in Equation 3, is preferable. Conversely, if the bias matrix is underperforming but still conveys positional information, the original implementation in Equation 2 is more effective.

**Multi-head DAPE**. In its simplest form, DAPE is considered for a single-head case as described in Equation 2 and Equation 3. However, adopting a multi-head mechanism significantly enhances model capabilities. To effectively combine both semantic and positional information, the DAPE in a multi-head setup processes the key-query similarities and bias matrices from all heads. Specifically, for an $h$-head layer, the function $f(\cdot)$ inputs a $2h$-dimensional concatenation of key-query similarities and positional biases. It then outputs $h$-dimensional vectors, where each element corresponds to the DAPE for the respective head. We have shown the code implementation in Appendix J. Importantly, semantic and positional information across different heads are processed simultaneously within the same MLPs, rather than sequentially. This approach not only improves computational efficiencies through parallel processing but also capitalizes on the richer semantic information available across all heads. Compared to the key-query similarity derived from a single head, the comprehensive attention from all heads yields more substantial semantic information.

**Computational costs analysis**. Here, we evaluate the additional computational costs introduced by the DAPE method, compared with the classical positional encoding methods (e.g., Alibi and Kerple). We consider a transformer model with $h$ heads and assume a sequence length of $N$ and all hidden dimensions in the attention layer being $d$. Then the total computational cost for a standard transformer equipped with classical PEs is $\mathcal{O}\left(hN^2d + hNd^2\right)$. When incorporating the proposed DAPE, which employs two-layer MLPs with hidden dimension $D_{\text{DAPE}}$, the additional computational costs are $\mathcal{O}\left(hN^2D_{\text{DAPE}}\right)$. If the hidden dimensions $D_{\text{DAPE}} \ll d$, the incremental computational cost introduced by DAPE is not significant.

# 4 Experiment

**Baselines.** We evaluate the proposed DAPE against a range of established baselines, including NoPE [33], RoPE [62], YaRN [51], Randomized RoPE [57, 31], T5's Bias [56], Alibi [52], Kerple [13], and FIRE [41]. For RoPE, the randomized positional encoding [57, 31] is applied to enhance the model performance, extending the randomized length to four times that of the training length.

**Datasets.** Our analysis involves training language models on the Arxiv and Books3 datasets, which are frequently used benchmarks for evaluating model performance [52, 13, 41, 24]. We start our evaluation by comparing the last 256 tokens' zero-shot perplexity across different input lengths. Besides perplexity as evaluation metrics, we also employ the downstream datasets in randomized positional encoding [57] to evaluate DAPE, where details are included in Appendix D.

**Experiment settings.** Initially, we compare DAPE with other baselines at training lengths of 128, 512, and 1024, with model size 125M decoder-only Transformers [9], whose configuration is shown in Appendix B. Subsequently, we evaluate the performance of larger model size 350M, DAPE variants and explore the impact of hidden dimension of MLPs $D_{\text{DAPE}}$. We also examine the computational efficiency of DAPE, focusing on processing times. Additionally, we provide visualizations of the DAPE bias in the Appendix I. Finally, we also evaluate DAPE on algorithmic reasoning datasets via accuracy metrics.

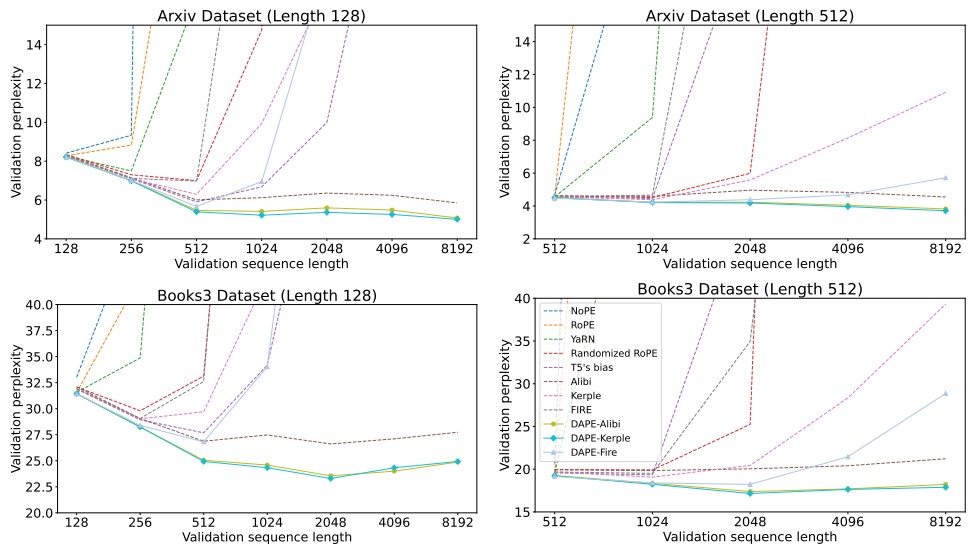

Figure 2: **Comparisons with baselines:** performance with training lengths 128 and 512 on Arxiv and Books3 datasets.

## 4.1 Comparisons with Baselines

**DAPE's superior performance within training length and beyond training length, compared to all baselines.** As shown in Figure 2 and Table 5, DAPE consistently outperforms established baselines such as RoPE, Alibi, and Kerple across various settings. Notably, DAPE-Kerple (the positional information in DAPE comes from Kerple bias matrices) outstands in both short and long training lengths (128 and 512), compared to previous RoPE, T5's bias, and so on. It demonstrates that the semantic adaptivity of DAPE significantly enhances its state-of-the-art performance against all other static positional encoding methods.

**The performance on longer training length 1024.** As shown in Figure 3, the proposed method consistently delivers state-of-the-art performance for the training length of 1024. When the evaluation extends to 2048, both DAPE-Kerple and DAPE-FIRE achieve notable results, recording performances of 3.91 and 3.93 perplexity scores, respectively. Remarkably, DAPE-FIRE behaves well at the longer evaluation length of 8192, achieving a performance of 3.91 scores and surpassing Alibi's score of

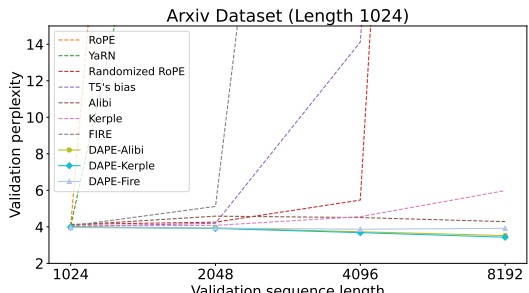

Figure 3: **Results on the training length 1024.**

4.28. These findings reveal that DAPE sustains robust performance with a longer training length of 1024.

**DAPE enhances intra-length performance, indicating that its lower perplexity may come from thorough utilization of entire sentences but not disregarding long-distance information (Also proved in Figure 1).** Compared to Alibi, Kerple, and FIRE, the adapted versions DAPE-Alibi, DAPE-Kerple, and DAPE-FIRE demonstrate consistently and significantly better intra-length performance. With the growing sequence length, the Alibi tends to transition from full attention to almost local attention, and this is why Alibi is worse than most baselines within training length but better beyond training lengths. The results (as shown in Table 5) indicate that the superior intra-length performance of DAPE is statistically significant, with a p-value less than 0.05. Therefore, the consistent intra-length performances across various training lengths indicate that the lower perplexity of DAPE results from effectively utilizing the entire sequence, rather than focusing on local parts and neglecting long-distance information.

**DAPE significantly improves length extrapolation performance, compared to ALibi, Kerple, and FIRE.** DAPE-Kerple significantly surpasses competitors like vanilla Kerple when training and evaluating at different lengths. On the Arxiv dataset trained at a length of 128, DAPE-Kerple achieves a remarkably low perplexity of 5.00 at an evaluation length of 8192, in stark contrast to Kerple's 31.93. Similarly, on the Books3 dataset with a training length of 512, DAPE-Kerple records a perplexity of 17.88 at the same extended evaluation length, far outperforming Kerple's 39.31. These results affirm that DAPE, through its semantic adaptivity and flexibility, consistently enhances performance beyond training lengths, eclipsing static positional encoding methods.

## 4.2 The Effect of Model Size

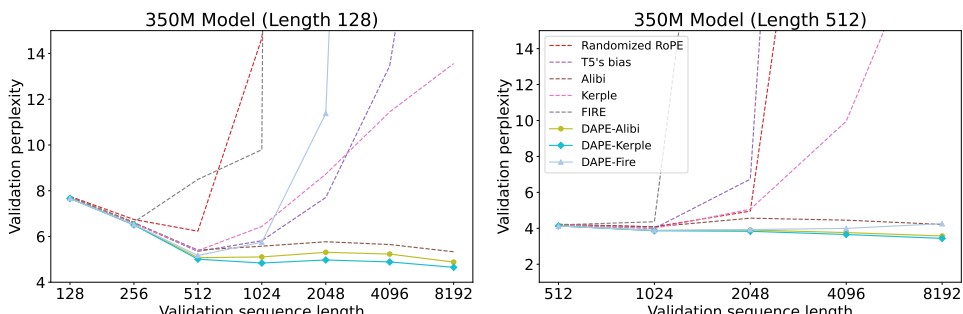

Figure 4: **The effect of model size:** for the 350M model, the performance with training lengths 128 and 512 on the Arxiv dataset.

**DAPE enhances performance with increasing model sizes.** As the model size grows (as shown in Figure 4), DAPE consistently demonstrates an improvement in performance metrics. When the model size is augmented from 125M to 350M, the perplexity at an evaluation sequence length of 8192 (with a training length of 512) for DAPE-Alibi shows a notable decrease from 3.82 to 3.57. These numbers are appreciably smaller than those recorded for original Alibi, which decreases from 4.54 to 4.21 in perplexity, indicating a robust performance improvement. Additionally, DAPE-Kerple significantly reduces the perplexity for Kerple, bringing it down from an initial 22.76 to an impressive 3.43. These results confirm that DAPE retains its efficacy and continues to perform well even as the model size is increased, mainly due to the adoption of semantically adaptive PEs.

**DAPE methods almost are ranked top-3 with large model size.** With the incremental model size, DAPE-FIRE begins to match, and nearly approach, the performance levels of Alibi. Initially, at a model size of 125M and a training length of 512, DAPE-FIRE achieves a perplexity of 5.71 at an evaluation sequence length of 8192, while Alibi stands at a perplexity of 4.54. However, as the model size is increased to 350M, the performance gap significantly narrows. Specifically, DAPE-FIRE outperforms Alibi regarding the perplexity scores when the evaluation length is smaller than 4096, as the model size grows for evaluation. In conclusion, as shown in Figure 4, we observe that the DAPE methods almost win the top-3 among all positional encoding methods. This trend underlines the scalability and adaptability of DAPE, emphasizing its potential to handle more substantial computation challenges.

## 4.3 Different Variants of DAPE

In this section, we evaluate the performance of DAPE across its various forms. Our analysis focuses on DAPE-Kerple. Notably, as shown in Figure 5, all variants of DAPE surpass the baseline performance of Kerple. The *Addition_Residual* variant of DAPE, while requiring less computational effort, delivers relatively inferior results. As illustrated in Figure 5, concatenation methods (either *Concat* or *Concat_Residual*) outperform the *Addition_Residual* approach, for both the training length of 128 and 512. Furthermore, both *Concat* and *Concat_Residual* exhibit comparable performance metrics. Specifically, at a training length of 128, *Concat_Residual* records a score of 5.00 and *Concat* scores 5.03 at an evaluation length of 8192, whereas *Add_Residual* posts a 5.17 perplexity score. With a training length of 512, *Concat_Residual* achieves a score of 3.70, and *Concat* scores 3.69 at an evaluation length of 8192, compared to *Add_Residual*'s 3.75. Based on the current observation, the different variants of DAPE show comparable performances, compared to baselines.

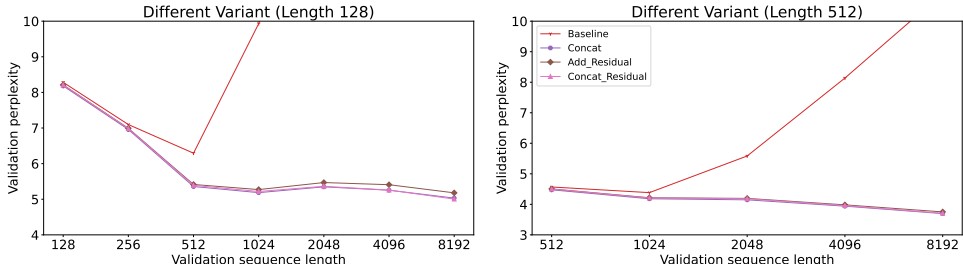

Figure 5: **Different variants of DAPE:** the DAPE-Kerple performance under different variants. (1) *Add_Residual*: $\boldsymbol{X}\boldsymbol{W}_Q(\boldsymbol{X}\boldsymbol{W}_K)^\top + \boldsymbol{B} + f(\boldsymbol{X}\boldsymbol{W}_Q(\boldsymbol{X}\boldsymbol{W}_K)^\top + \boldsymbol{B})$; (2) *Concate*: $\boldsymbol{X}\boldsymbol{W}_Q(\boldsymbol{X}\boldsymbol{W}_K)^\top + f(\boldsymbol{X}\boldsymbol{W}_Q(\boldsymbol{X}\boldsymbol{W}_K)^\top, \boldsymbol{B})$; (3) *Concate_Residual*: $\boldsymbol{X}\boldsymbol{W}_Q(\boldsymbol{X}\boldsymbol{W}_K)^\top + \boldsymbol{B} + f(\boldsymbol{X}\boldsymbol{W}_Q(\boldsymbol{X}\boldsymbol{W}_K)^\top, \boldsymbol{B})$.

## 4.4 The Effect of the Hidden Dimension $D_{\textbf{DAPE}}$

**Even small $D_{\textbf{DAPE}}$ can improve the performance.**   The experiments are conducted with Alibi and DAPE-Alibi. As shown in Appendix Figure 6, when considering the training length 128 and $D_{\text{DAPE}}$ is set as 4, the DAPE-Alibi achieves 8.25 at evaluation length 128 and 5.67 at length 8192, which is better than Alibi's 8.31 and 5.85. Whatever $D_{\text{DAPE}}$ is 4, 16 32, or 64, the performance is always better than the original Alibi at all evaluation lengths. This suggests the effectiveness of DAPE, even with smaller $D_{\text{DAPE}}$.

**The choice of $D_{\textbf{DAPE}}$.**   Based on the experiment, overly small values of $D_{\text{DAPE}}$ can degrade performance, although they still perform better than the baseline. Conversely, larger values of $D_{\text{DAPE}}$ increase computational costs. The function $f(\cdot)$ is implemented as a two-layer MLP, where the input dimension is either the head number or twice the head number, and the output dimension is the head number. Therefore, we recommend setting the hidden dimension to the head number to prevent information loss and ensure the capacity of $f(\cdot)$.

## 4.5 The Time Cost

Table 1: The time cost (millisecond) under different testing lengths, with $D_{\textbf{DAPE}}$ as 32 and default batch size 1, with training length 512.

| Method | 350M Total | Ratio | 2.7B Total | Ratio | 6.7B Total | Ratio |
|---|---|---|---|---|---|---|
| RoPE [62] | 210.01 | 0.9366 | 472.63 | 1.1187 | 635.57 | 0.8858 |
| T5's bias [56] | 355.16 | 1.5839 | 537.62 | 1.2725 | 808.85 | 1.1273 |
| ALiBi [52] | 172.60 | 0.7697 | 325.95 | 0.7715 | 596.77 | 0.8317 |
| Kerple [13] | 189.91 | 0.8469 | 370.32 | 0.8765 | 661.82 | 0.9224 |
| FIRE [41] | 248.13 | 1.1066 | 432.63 | 1.0240 | 797.68 | 1.1118 |
| DAPE-Kerple | 224.22 | 1.0000 | 422.48 | 1.0000 | 717.46 | 1.0000 |

**Practical additional time cost.**   The additional training ratio will gradually decrease with a larger model size, compared to baseline Kerple.. The cost of Feed-Forward Network is: $O(Nd_{head}^2 d_{hidden}^2)=aNd_{head}^2 d_{hidden}^2$, where a is a constant, N is the sequence length, $d_{head}$ is the attention head number and $d_{hidden}$ is the dimension for attention calculation. The cost of Attention: $O(N^2 d_{head} d_{hidden})=bN^2 d_{head} d_{hidden}$, where b is a constant. The additional cost of DAPE: $O(N^2 d_{head} d_{DAPE})=cN^2 d_{head} d_{DAPE}$, where c is a constant. The cost ratio is $\frac{aNd_{head}^2 d_{hidden}^2+bN^2 d_{head} d_{hidden}}{aNd_{head}^2 d_{hidden}^2+bN^2 d_{head} d_{hidden}+cN^2 d_{head} d_{DAPE}}=\frac{ad_{head} d_{hidden}^2+bNd_{hidden}}{ad_{head} d_{hidden}^2+bNd_{hidden}+cNd_{DAPE}}$. Therefore, with the fixed sequence length and $d_{DAPE}$, with the model becomes larger (with bigger $d_{head}$ and $d_{hidden}$), the additional cost ratio of DAPE will greatly become smaller. Also, we have shown in Figure 6 that $d_{DAPE}$ still works well with very small value, such as 4.

Table 2: Train on length 40 with 200k steps, and test from lengths 41 to 500. The random accuracy is 50%, except for MODULAR ARITHMETIC (SIMPLE), CYCLE NAVIGATION, BUCKET SORT, SOLVE EQUATION and MODULAR ARITHMETIC, where it is 20%. ††† denotes permutation-invariant tasks, which are expected to be solved without positional information.

| Level | Task | Randomized | | | | | | | DAPE (Ours) | | |
|---|---|---|---|---|---|---|---|---|---|---|---|
| | | Learned | sin / cos | RoPE | Relative [19] | ALiBi | Kerple | FIRE | Alibi | Kerple | FIRE |
| R | EVEN PAIRS | 50.04 | 91.27 | 99.98 | 96.60 | 73.52 | 57.50 | 73.86 | 99.99 | 99.58 | **100** |
| | MODULAR ARITHMETIC (SIMPLE) | 19.95 | 20.39 | 21.35 | 20.84 | 20.02 | 21.79 | 21.09 | 23.58 | **24.47** | 24.46 |
| | PARITY CHECK††† | 50.14 | 50.52 | 50.05 | 50.09 | 50.09 | 50.07 | **50.97** | 50.30 | 50.07 | 50.04 |
| | CYCLE NAVIGATION††† | 24.97 | 25.37 | 27.63 | 26.95 | 24.64 | 29.47 | 28.41 | 22.99 | **34.53** | 27.54 |
| DCF | STACK MANIPULATION | 59.92 | 65.92 | 61.49 | 64.73 | 66.42 | 66.93 | 69.33 | 68.18 | **72.04** | 70.90 |
| | REVERSE STRING | 52.76 | 67.28 | 65.23 | 65.59 | 71.09 | 71.54 | 65.89 | 73.37 | 70.74 | **76.40** |
| | MODULAR ARITHMETIC | 31.00 | 30.70 | 31.25 | 31.74 | 30.56 | 24.79 | 30.92 | 31.34 | **32.37** | 31.50 |
| | SOLVE EQUATION | 20.00 | 19.97 | 21.85 | **22.93** | 19.92 | 21.15 | 22.06 | 20.03 | 22.49 | 22.42 |
| CS | DUPLICATE STRING | 52.77 | 65.44 | 64.97 | 67.66 | 65.13 | 66.72 | 69.03 | 70.84 | **72.95** | 72.71 |
| | MISSING DUPLICATE | 50.38 | 49.78 | 63.37 | 72.34 | 74.21 | 79.06 | 79.27 | 83.41 | 87.57 | **89.17** |
| | ODDS FIRST | 52.77 | 58.61 | 61.00 | 61.57 | 59.88 | 62.59 | 63.28 | 63.78 | **67.08** | 66.34 |
| | BINARY ADDITION | 54.63 | 55.78 | 55.59 | 56.96 | 54.72 | 56.35 | 55.70 | 59.71 | **60.88** | 56.62 |
| | COMPUTE SQRT | 50.47 | 51.11 | 51.88 | 51.63 | 50.63 | 51.11 | 50.80 | 51.64 | 51.33 | **52.46** |
| | BUCKET SORT††† | 98.32 | 98.92 | 98.12 | 99.31 | 98.45 | 99.38 | **99.57** | 99.38 | 98.81 | 99.37 |

## 4.6 The Visualization of DAPE

In this subsection, we present the visualization of learned positional encoding biases from a DAPE-Kerple model pretrained on Arxiv (training length is 512). We plot the learned positional encoding bias for the query token at the 8192th position, for all the attention heads from selected layers in Figure 1. We would like to highlight two features of DAPE. First, in different attention heads, the bias matrix of DAPE learns both local and "anti-local" attention patterns that emphasize more on far-away keys (just like FIRE), compared to a fixed local inductive bias (such as Kerple and Alibi). Secondly, the bias matrix can be dynamically adjusted with different attention values, compared to the static bias fixed for all attentions. We have shown more examples, including different layers and different samples, in Appendix I.

## 4.7 Experiments on CHE Benchmark

Besides employing perplexity as an evaluation metric, we also evaluated DAPE on downstream Chomsky Hierarchy Evaluation Benchmark (CHE) [21] (need to utilize the whole sentence information to generate correct answers) to further discuss its effects. The experimental setup follows randomized positional encodings [57], detailed in Table 4, with the experiment setting shown in Appendix Section D. Overall, FIRE outperforms Kerple in 9 out of 14 tasks, while Kerple outperforms Alibi in 11 out of 14 tasks. This observation aligns with findings in [41], suggesting that the experiments in Table 2 are reliable and reflect the performance of positional encoding in downstream tasks.

**DAPE works better on permutation-variant tasks.** DAPE (with Kerple and FIRE) presented the best performance in 10 out of 11 permutation-variant tasks (which require positional information), achieving the second-best performance in the SOLVE EQUATION task. This underscores the efficacy of DAPE with semantic adaptivity in handling permutation-variant challenges.

**DAPE's performance on permutation-invariant tasks.** In tasks that are permutation-invariant, where positional information is non-critical, DAPE demonstrated comparable performance. Notably, DAPE-Alibi achieved scores of 50.30 on PARITY CHECK and 99.38 on BUCKET SORT tasks, compared to the highest scores of 50.97 and 99.57, respectively, demonstrating competitive performances.

**Comparative performance improvements.** DAPE consistently enhanced performance across various tasks, especially on permutation-variant tasks. Specifically, DAPE improved upon Alibi and FIRE's results in all 11 tested permutation-invariant tasks. Similarly, it outperformed Kerple in 10 of these tasks. These results highlight the effectiveness of DAPE over static positional encoding methods like Alibi, Kerple, and FIRE, resulting from its dynamic adaptivity.

# 5   Evaluation Protocol

In this work, we initially utilize the model to process the entire input sentence and subsequently select the final 256 tokens for perplexity computation. This approach contrasts with a variety of other studies, which employ methods that process the full sequence during perplexity calculations [51, 31, 12, 24, 10, 41]. As a result, our reported baseline perplexity is comparatively higher than the results presented in ALiBi [52], which adopt a non-overlapping evaluation strategy. This method divides sequences longer than $L$ into multiple segments of length $L$, thereby yielding lower perplexity figures.

Though ALiBi [52] and Kerple all claim that they use non-overlapping evaluations, their reported results are different. For example, in the ALiBi paper Table 2, the sinusoidal position encoding perplexity increases from 20.05 (evaluation length 512) to 406.01 (evaluation length (evaluation length 15512), while in Kerple paper Table 3 the sinusoidal position encoding perplexity from 33 to 30046. This may be caused by **different evaluation protocols and training strategies.**

**Recommended Protocols.**   We strongly recommend that researchers process the entirety of the sequence before selecting the last $K$ tokens for the purpose of calculating perplexity. The rationale behind processing the whole sentence is that it provides a comprehensive evaluation of the model's capability to handle long-context dependencies, thus offering a more accurate reflection of its performance. Following this step, we advocate for the selection of the last $K$ tokens to compute perplexity, ensuring that the same number of tokens is used across different evaluation lengths, which promotes consistency and comparability in the results.

**Release this work's code for future work.**   In light of this methodology, we have made our code publicly available to other researchers in the field. This initiative aims to facilitate a standardized comparison and evaluation of their respective methods, thereby advancing the collective understanding of model performance in relation to perplexity calculations.

# 6   Conclusion

In this paper, we propose the data-adaptive positional encoding (DAPE) by incorporating both the semantic and the positional information to improve the model performance. We show that the additional computation introduced by DAPE is not significant under proper choices of hyperparameters. We conduct comprehensive experiments on Arxiv, Books3, and CHE to validate the effectiveness of the proposed method, revealing that the adaptive PE method has advantages over static PEs. We believe that the DAPE could benefit the whole community, especially on length generalization tasks.

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

# A  Broader impacts

**Positive societal impacts.**  We propose a method for length extrapolation, which will be helpful for transformer-based models to process long context.

**Negative societal impacts.**  This method may be abused for other potential long-context applications.

# B  Model Configuration

All experiments are conducted on 8 x A800 GPUs. The 125M model configuration is the following.

Table 3: **Model Configurations.**

|  | 125M | 350M |
| --- | --- | --- |
| Training sequence length | 512 | 512 |
| Batch size | $32 \times 8$ | $32 \times 8$ |
| Numer of iterations | 50k | 50k |
| Dropout prob. | 0.0 | 0.0 |
| Attention dropout prob. | 0.0 | 0.0 |
| Attention head | 12 | 16 |
| Feature dimension | 768 | 1024 |
| Layer number | 12 | 24 |
| Optimizer | Adam | Adam |
| Optimizer parameter betas | [0.9, 0.95] | [0.9, 0.95] |
| Learning rate | $6e - 4$ | $3e - 4$ |
| Precision | float16 | float16 |

## C    Appendix: The Effect of the Hidden Dimension $D_{\mathbf{DAPE}}$

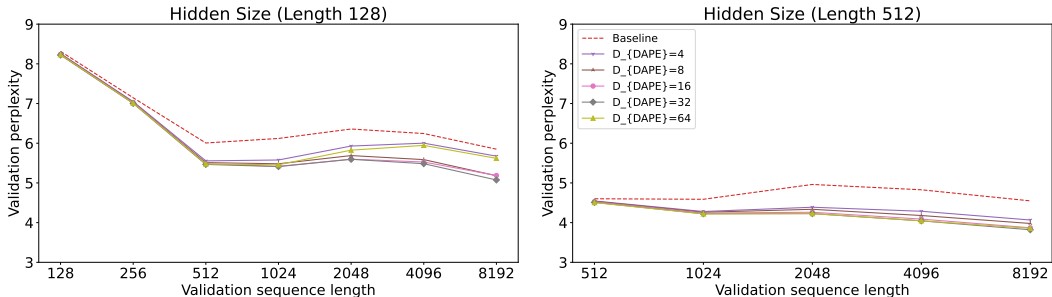

Figure 6: **The effect of $D_{\mathbf{DAPE}}$: the performance with training lengths 128 and 512 on the Arxiv dataset. The experiments are conducted with Alibi and DAPE-Alibi.**

## D    Experiments on Chomsky Hierarchy Evaluation Benchmark (CHE)

Following the framework established by [21, 57], we conduct evaluations of our DAPE on a suite of tasks derived from the domain of formal language recognition. These tasks include modular arithmetic, reversing and duplicating strings, binary operations and bucket sort. Based on the Chomsky hierarchy [16], these tasks are categorized into distinct classes: Regular (R), Context-Free, Context-Sensitive (CS), and Recursively Enumerable. Each class aligns with specific computational models: Regular tasks are solvable using Finite-State Automata (FSA); Deterministic Context-Free tasks can be addressed by an FSA equipped with a deterministic stack; and Context-Sensitive tasks require an FSA complemented by access to a bounded tape.

Table 4: The example of different tasks. ††† denotes permutation-invariant tasks, which are expected to be solved without positional information. More explanation of tasks can be found in [21].

| Level | Task | Example Input | Example Output |
|---|---|---|---|
| Regular | EVEN PAIRS | $aabba$ | True |
| | MODULAR ARITHMETIC (SIMPLE) | $1 + 2 - 4$ | 4 |
| | PARITY CHECK††† | $aaabba$ | True |
| | CYCLE NAVIGATION††† | $011210$ | 2 |
| DCF | STACK MANIPULATION | $abbaa$ POP PUSH $a$ POP | $abba$ |
| | REVERSE STRING | $aabba$ | $abbaa$ |
| | MODULAR ARITHMETIC | $-(1 - 2) \cdot (4 - 3 \cdot (-2))$ | 0 |
| | SOLVE EQUATION | $-(x - 2) \cdot (4 - 3 \cdot (-2))$ | 1 |
| CS | DUPLICATE STRING | $abaab$ | $abaababaab$ |
| | MISSING DUPLICATE | $10011021$ | 0 |
| | ODDS FIRST | $aaabaa$ | $aaaaba$ |
| | BINARY ADDITION | $10010 + 101$ | 10111 |
| | COMPUTE SQRT | $100010$ | 110 |
| | BUCKET SORT††† | $421302214$ | 011222344 |

**Problem Setting**    Building upon the framework proposed by Ruoss et al. (2023) [57], we utilize the encoder-only configuration of the original sequence-to-sequence Transformer model, as delineated by Vaswani et al. (2017) [69]. In scenarios that necessitate a multi-token output sequence $y$, such as the task of string duplication, we extend the input sequence by appending $|y|$ placeholder tokens. Subsequently, we compute the entire Transformer output from this augmented sequence without resorting to autoregressive sampling techniques. Training is conducted on sequences whose lengths are uniformly distributed, sampled from $U(1, N)$, with $N$ set to 40. Evaluation is performed on sequences that vary in length from $N + 1$ to $M$, where $M$ equals 500. The implemented architecture comprises 5 layers, 8 attention heads, and a feature dimension of 256. The dimension $D_{\text{DAPE}}$ is established at 64, while the maximum randomized position $L$ is set to 2048.

# E The Error Bar and Significance Value

Table 5: **The perplexity performances on the Arxiv dataset when the training length is 512 and running with three random seeds.**

| Method | | 512 | 1024 | 2048 | 4096 | 8192 |
|---|---|---|---|---|---|---|
| RoPE | mean | 4.5755 | 45.1974 | 134.1615 | 222.3333 | 265.4545 |
| | std | 0.0216 | 12.2241 | 21.8522 | 22.5955 | 20.3862 |
| T5's bias | mean | 4.5547 | 4.3730 | 12.1017 | 163.9289 | 595.0829 |
| | std | 0.0204 | 0.0747 | 6.5327 | 136.8294 | 306.3671 |
| Alibi | mean | 4.6146 | 4.5475 | 4.8693 | 5.0278 | 4.7679 |
| | std | 0.0278 | 0.0990 | 0.1246 | 0.1845 | 0.2196 |
| DAPE-Alibi | mean | 4.5262 | 4.1921 | 4.1068 | 4.1315 | 4.0013 |
| | std | 0.0270 | 0.0653 | 0.0885 | 0.0617 | 0.2708 |
| Kerple | mean | 4.5817 | 4.3504 | 5.4438 | 8.7511 | 13.3524 |
| | std | 0.0252 | 0.0526 | 0.3187 | 1.1066 | 2.4109 |
| DAPE-Kerple | mean | 4.5123 | 4.1716 | 4.0505 | 4.0033 | 3.8642 |
| | std | 0.0251 | 0.0637 | 0.0973 | 0.0333 | 0.2342 |
| FIRE | mean | 4.5741 | 4.6953 | 30.1164 | 165.5394 | 308.6173 |
| | std | 0.0227 | 0.1125 | 4.4317 | 41.2065 | 78.4652 |
| DAPE-FIRE | mean | 4.4879 | 4.1990 | 4.2826 | 4.7983 | 6.1623 |
| | std | 0.0206 | 0.0619 | 0.0709 | 0.1701 | 0.9226 |

According to Table 5, the performances of different methods are evaluated based on perplexity across various validation lengths ranging from 512 to 8192. The results indicate that DAPE-Alibi consistently outperforms the Alibi method, DAPE-Kerple surpasses Kerple, and DAPE-FIRE shows better performance than FIRE, all with p-values less than 0.05, suggesting significant improvements.

DAPE-Alibi demonstrates lower perplexity values compared to Alibi at all lengths, indicating more effective learning and generalization. Similarly, DAPE-Kerple shows significant improvements over Kerple, especially at larger lengths, where the gap in performance widens. The DAPE-FIRE method also shows notable enhancements over the FIRE method, particularly at higher lengths where standard FIRE struggles with increased perplexity.

Moreover, DAPE-Alibi, DAPE-Kerple, and DAPE-FIRE not only perform better than their respective original methods but also show superiority over RoPE and T5's bias across all validation lengths. This suggests that the modifications implemented in DAPE versions provide a more robust and generalized model, capable of maintaining lower perplexity and thus better performance on the Arxiv dataset.

In conclusion, the statistical analysis confirms that the proposed DAPE variations offer significant improvements in perplexity performance, thereby validating their effectiveness in comparison to their original counterparts and other baseline methods.

# F Data-Adaptive Related Position Encoding Performance Comparison

Table 6: The performance comparison between data-related position encoding, with dataset Books3 and training length 128.

| Method | 128 | 256 | 512 | 1024 | 2048 | 4096 | 8192 |
|---|---|---|---|---|---|---|---|
| Transformer-XL | 31.57 | 28.49 | 26.07 | 26.98 | 27.90 | 32.76 | 41.12 |
| CoPE | 31.61 | 28.41 | 25.79 | 27.96 | 33.80 | 54.08 | 90.66 |
| DAPE-Kerple | 31.49 | 28.27 | 24.93 | 24.31 | 23.34 | 24.38 | 25.01 |

According to the experiment, the transformer-xl achieves good performance on length extrapolation. Therefore, this also suggests that the position encoding should interact with attention/query/key to further improve the performance.

## G    Ablation Study on Bias Matrix

Table 7: Books Dataset Results: Train Length 512

|  | 512 | 1024 | 2048 | 4096 | 8192 |
|---|---|---|---|---|---|
| $QK^T + B$ (baseline) | 19.68 | 19.06 | 20.44 | 28.34 | 39.31 |
| $QK^T + B + f(B)$ | 19.64 | 18.83 | 18.49 | 20.62 | 23.49 |
| $QK^T + B + f(QK^T, B)$ | 19.22 | 18.22 | 17.15 | 17.63 | 17.88 |

We further conduct an ablation study on the $f$, proving that $f$ help enhance the bias matrix. DAPE improves the bias matrix for $A_{final}$, while the $A_{final}$ is used to calculate $A_{final}K$. For the unseen position, the $B$ partially could handle it (FIRE changes the problem to interpolation), but is not accurate enough so that DAPE helps enhance the bias matrix $B$ via attention score. The experiment suggests two points: 1) The DAPE $f(QK^T, B)$ is better than naive $f(B)$, suggesting that the context-adaptive is important. 2) The $QK^T + B + f(B)$ is better than $QK^T + B$, suggesting that benefiting from improving the expressiveness of the bias matrix.

## H    The result on 2.7B and 6.7B

Table 8: Model Size and Method Comparison with Training Length 512

| Model Size | Method | 512 | 1024 | 2048 | 4096 |
|---|---|---|---|---|---|
| 2.7B | RoPE | 21.01 | 25.00 | 48.13 | 160.59 |
|  | RPE | 21.10 | 21.88 | 23.59 | 33.23 |
|  | Alibi | 21.23 | 22.17 | 22.91 | 23.22 |
|  | Kerple | 21.14 | 22.08 | 23.38 | 27.21 |
|  | DAPE-Kerple | 20.52 | 21.01 | 20.23 | 19.57 |

Table 9: Model Size and Method Comparison with Training Length 512

| Model Size | Method | 512 | 1024 | 2048 |
|---|---|---|---|---|
| 6.7B | RoPE | 20.86 | 22.27 | 28.01 |
|  | RPE | 20.79 | 21.60 | 22.32 |
|  | Alibi | 20.79 | 21.63 | 22.45 |
|  | Kerple | 20.71 | 21.57 | 22.07 |
|  | DAPE-Kerple | 20.09 | 20.54 | 19.83 |

According to the result, we can find that the proposed DAPE still works well, whatever the model size is 2.7B or 6.7B. With 2.7B model size, RoPE achieves 21.01 on evaluation length 512 and 160.50 on evaluation 4096, while our DAPE-Kerple achieves 20.52 and 19.57 respectively. Also, DAPE-Kerple achieves the best performance, whatever the model size is 2.7B or 6.7B from evaluation length 512 to 8192. This suggests that our proposed DAPE has great scalability.

# I DAPE Visualization

The model is trained with DAPE-Kerple on length 512.

## I.1 Visualization on length 512

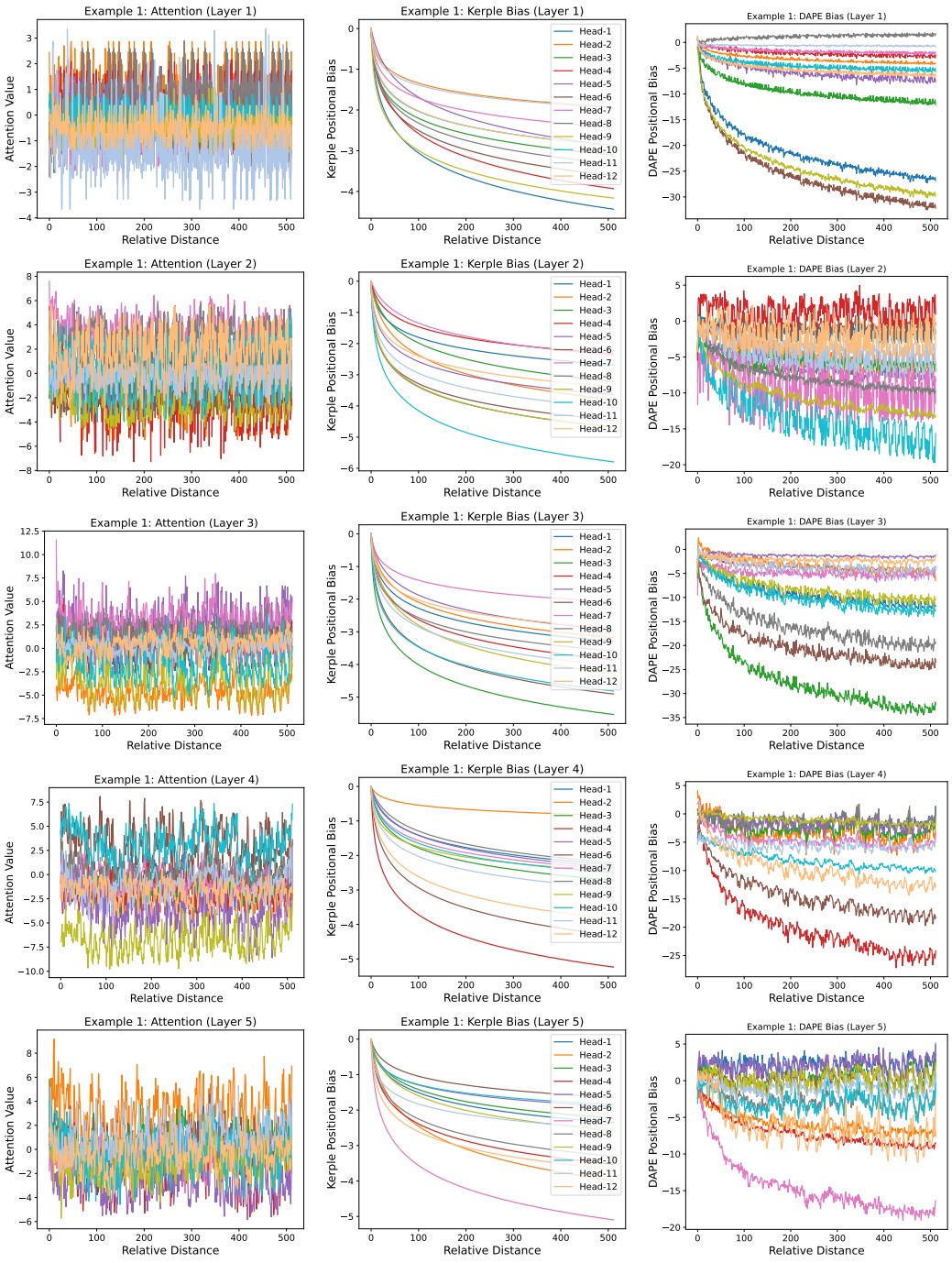

Figure 7: **Evaluation Length 512 Example 1: Part 1**

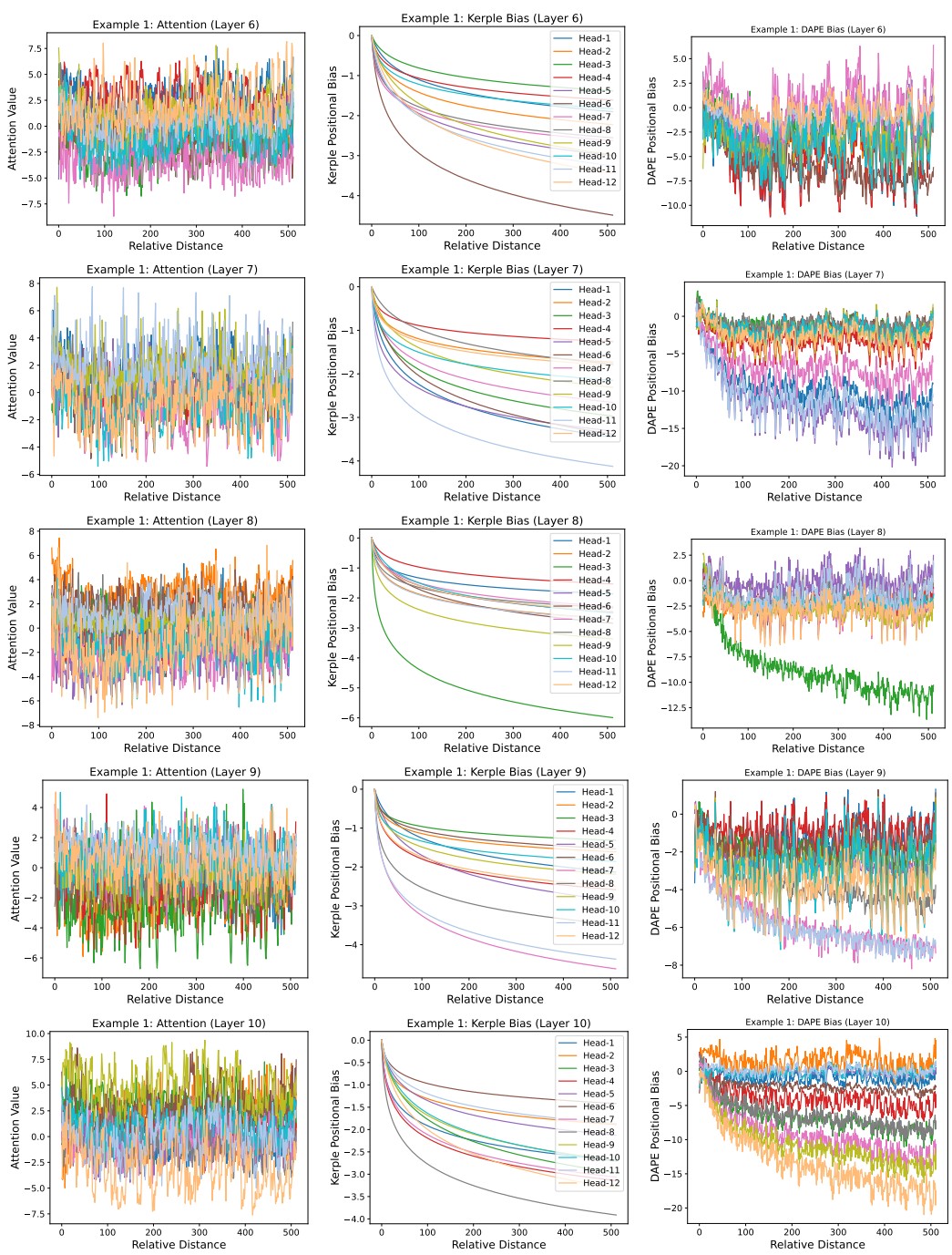

Figure 8: **Evaluation Length 512 Example 1: Part 2**

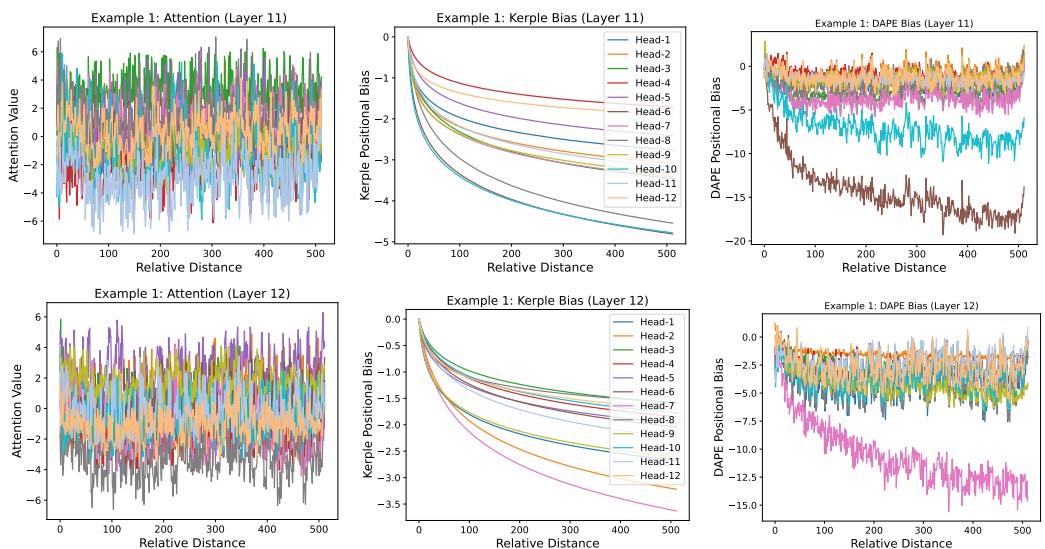

Figure 9: **Evaluation Length 512 Example 1: Part 3**

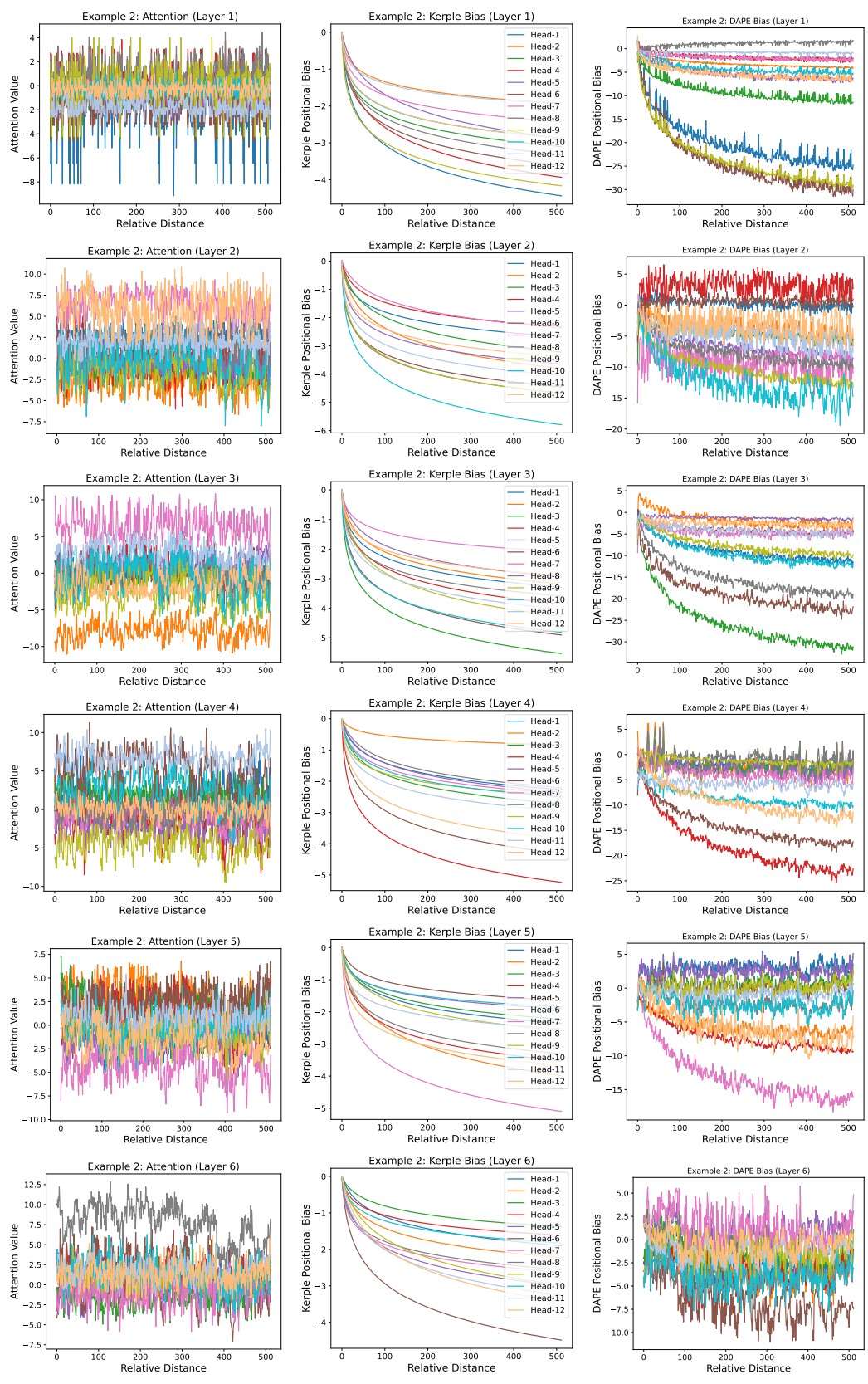

Figure 10: **Evaluation Length 512 Example 2: Part 1**

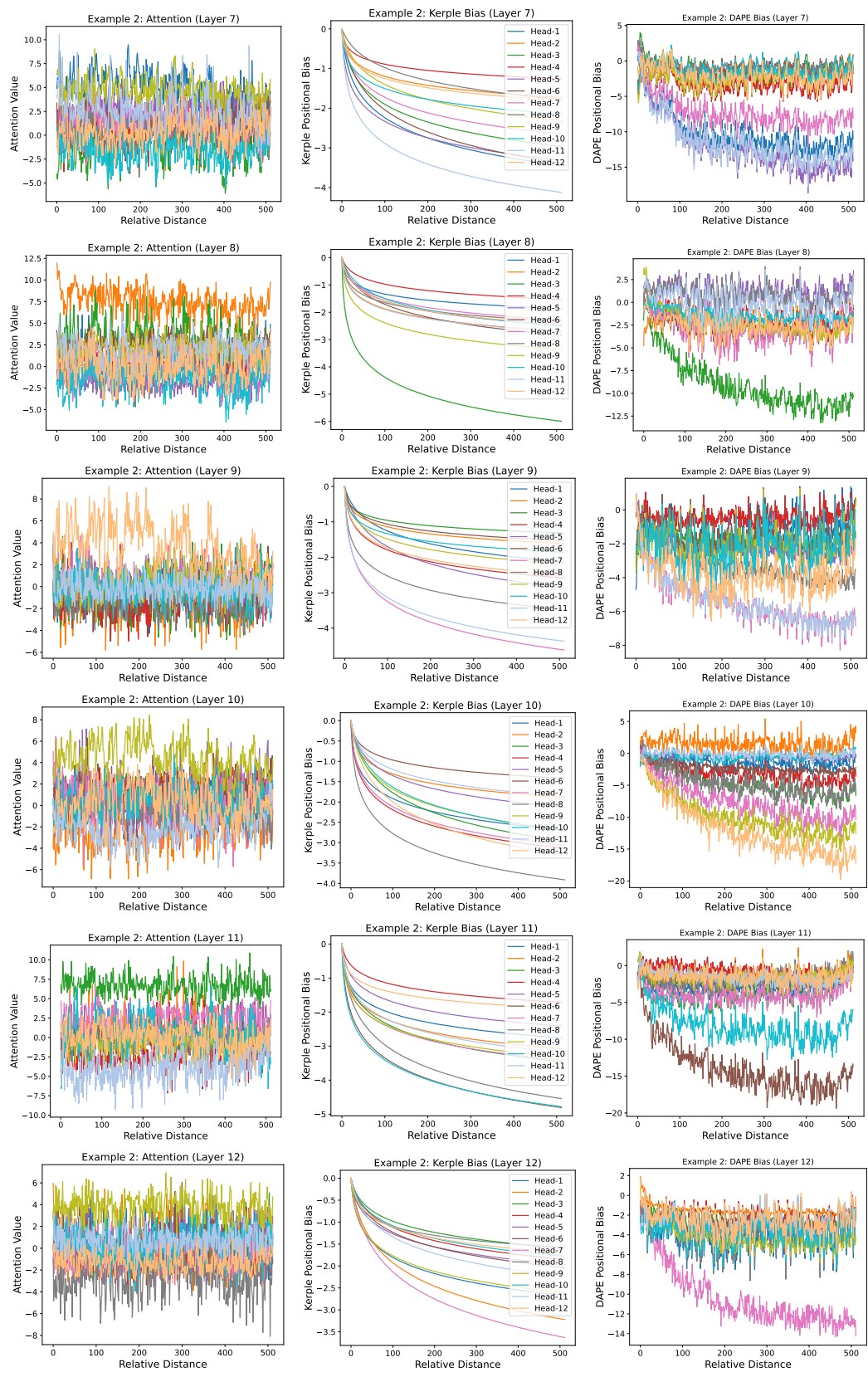

Figure 11: **Evaluation Length 512 Example 2: Part 2**

## I.2 Visualization on length 2048

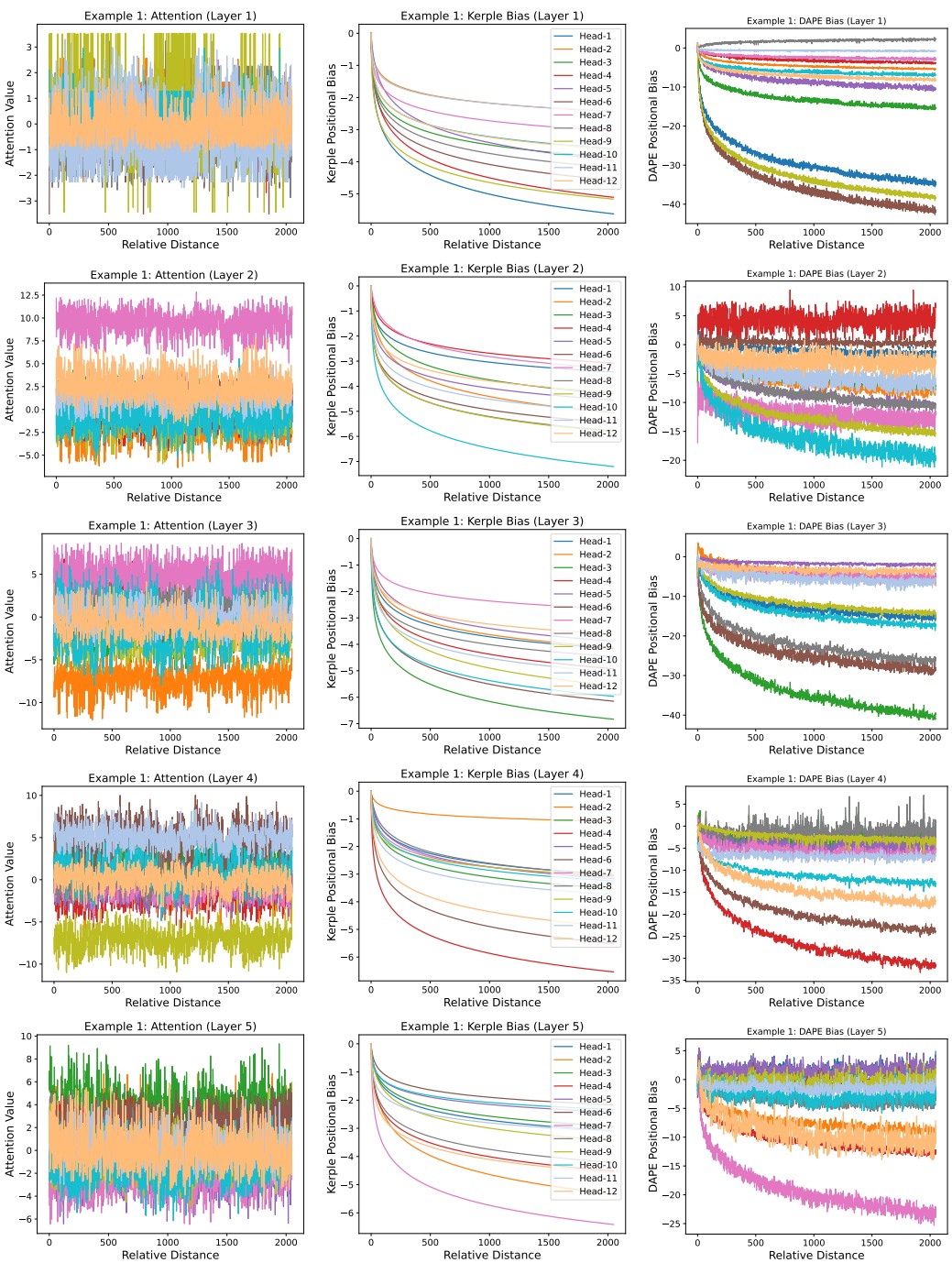

Figure 12: **Evaluation Length 2048 Example 1: Part 1**

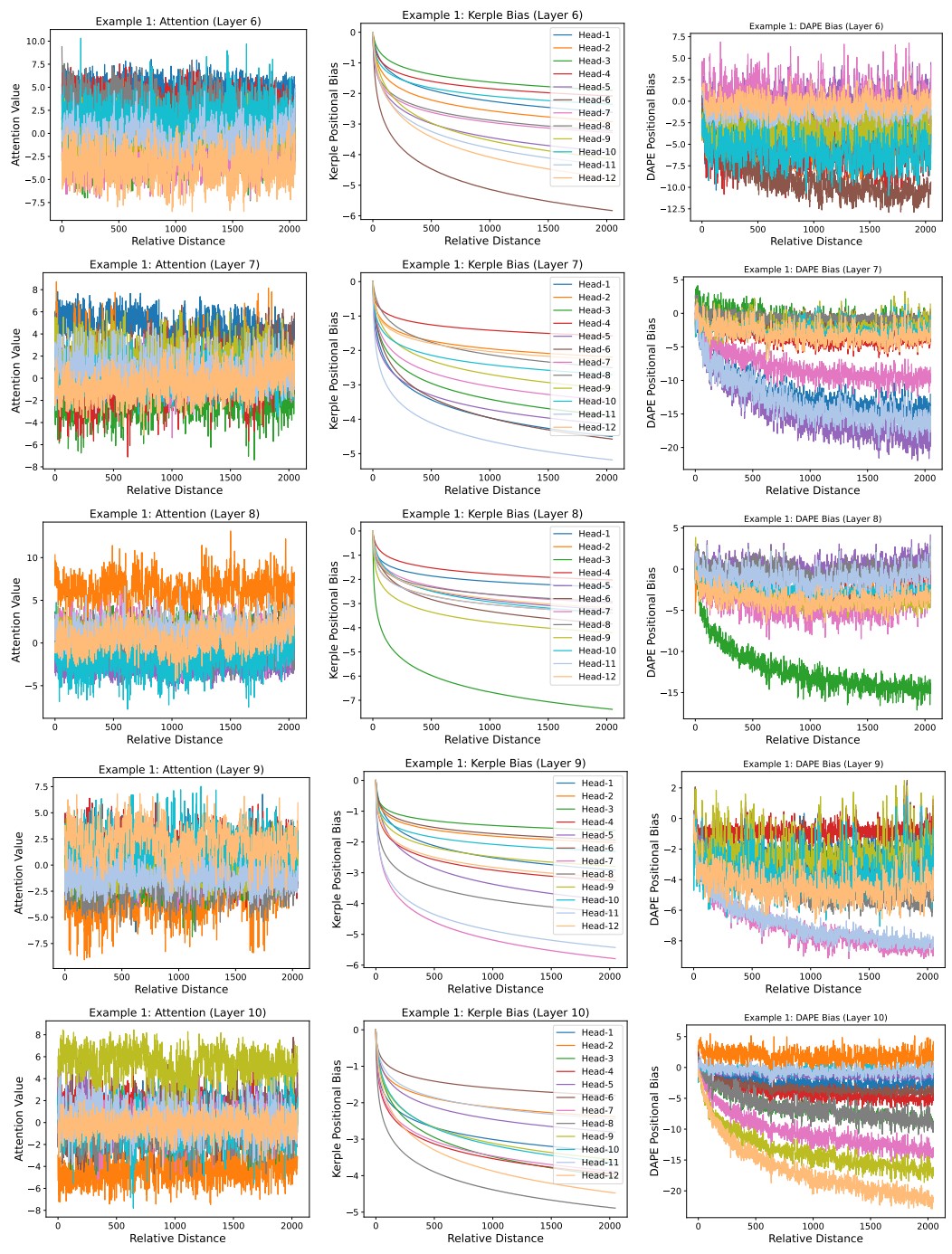

Figure 13: **Evaluation Length 2048 Example 1: Part 2**

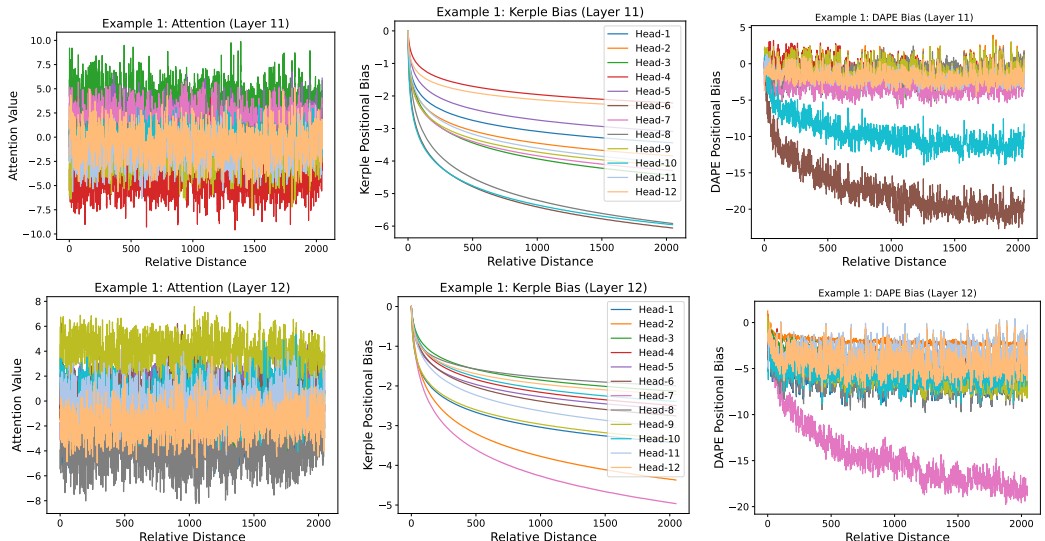

Figure 14: **Evaluation Length 2048 Example 1: Part 3**

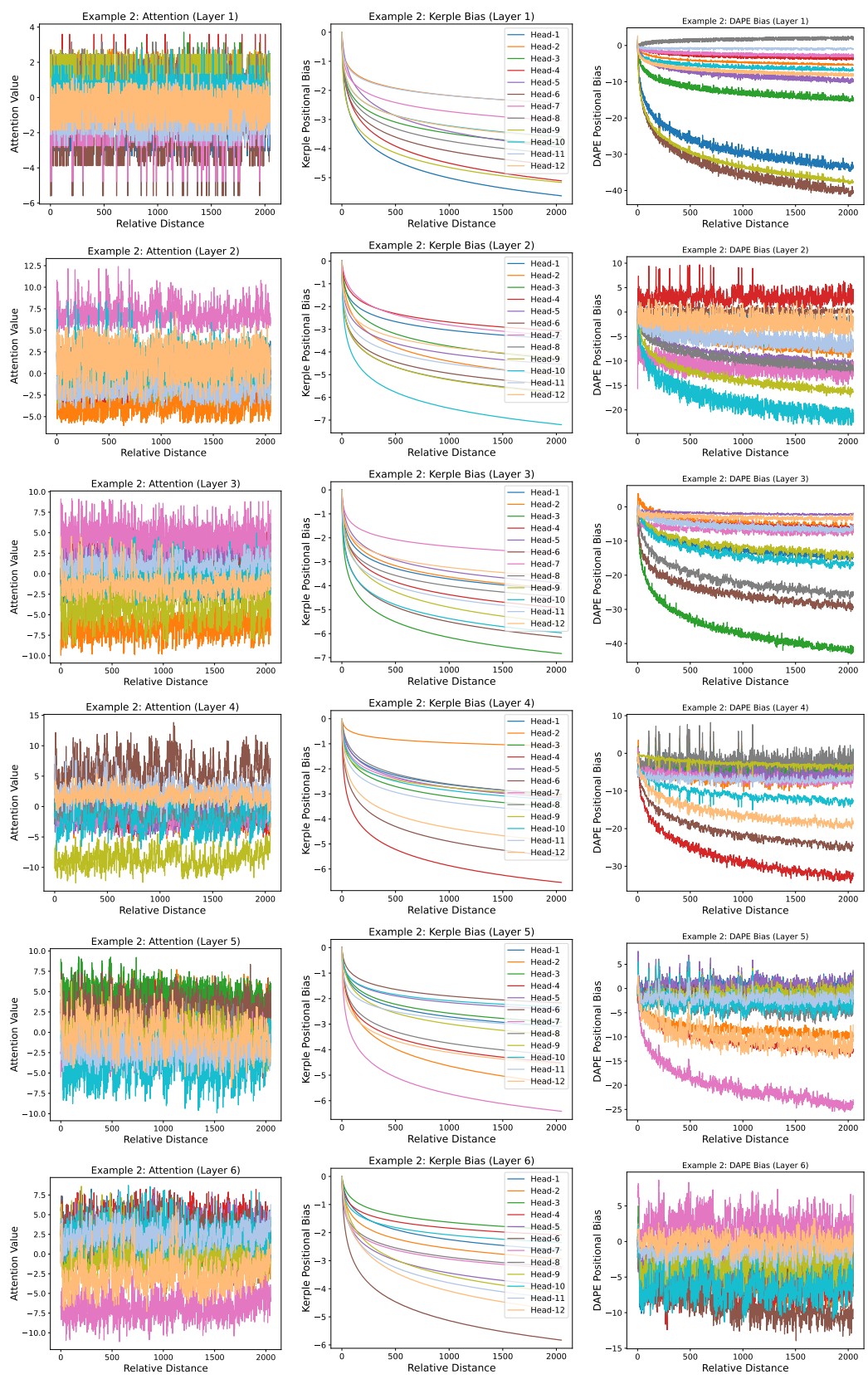

Figure 15: **Evaluation Length 2048 Example 2: Part 1**

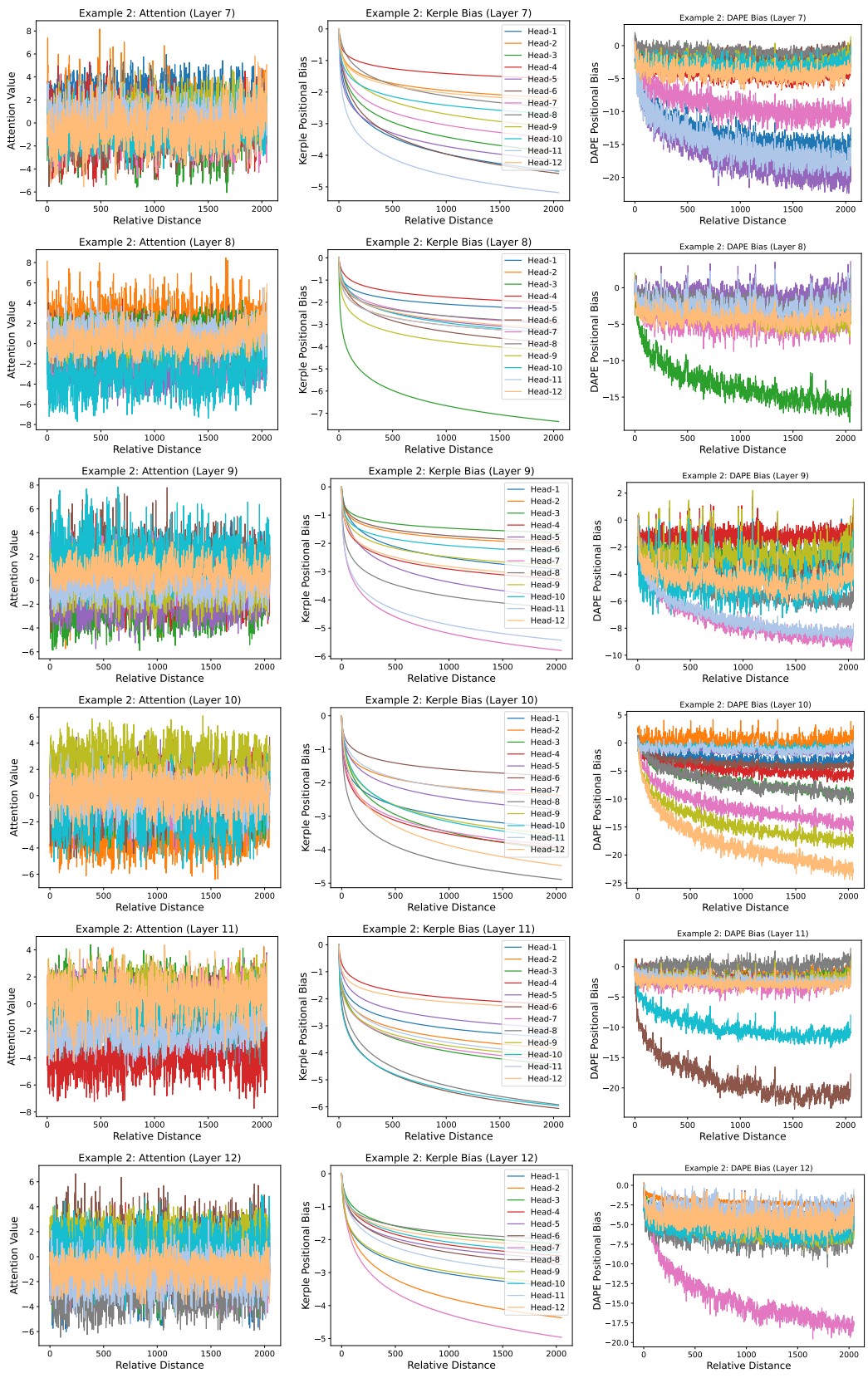

Figure 16: **Evaluation Length 2048 Example 2: Part 2**

## I.3 Visualization on length 8192

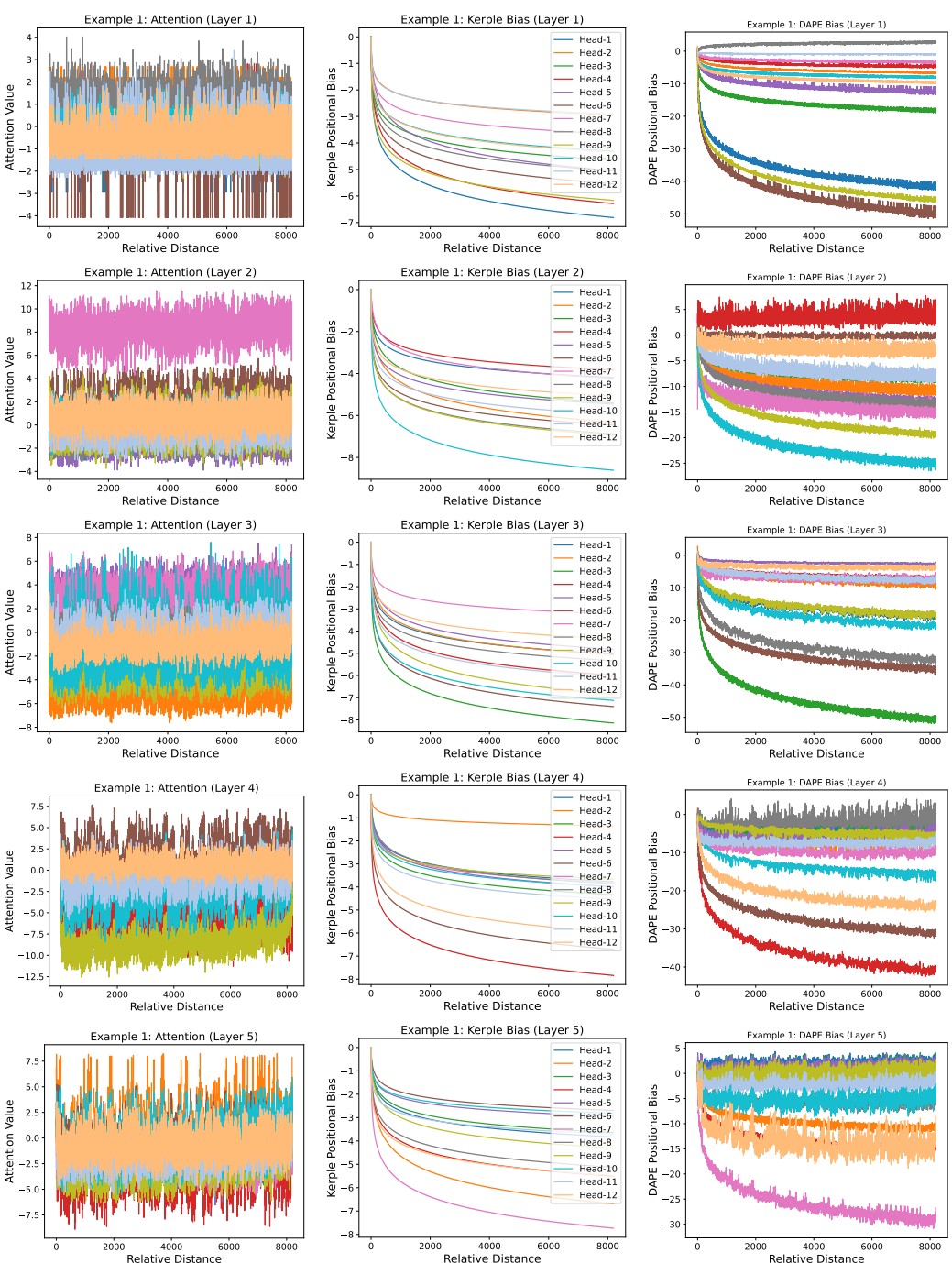

Figure 17: **Evaluation Length 8192 Example 1: Part 1**

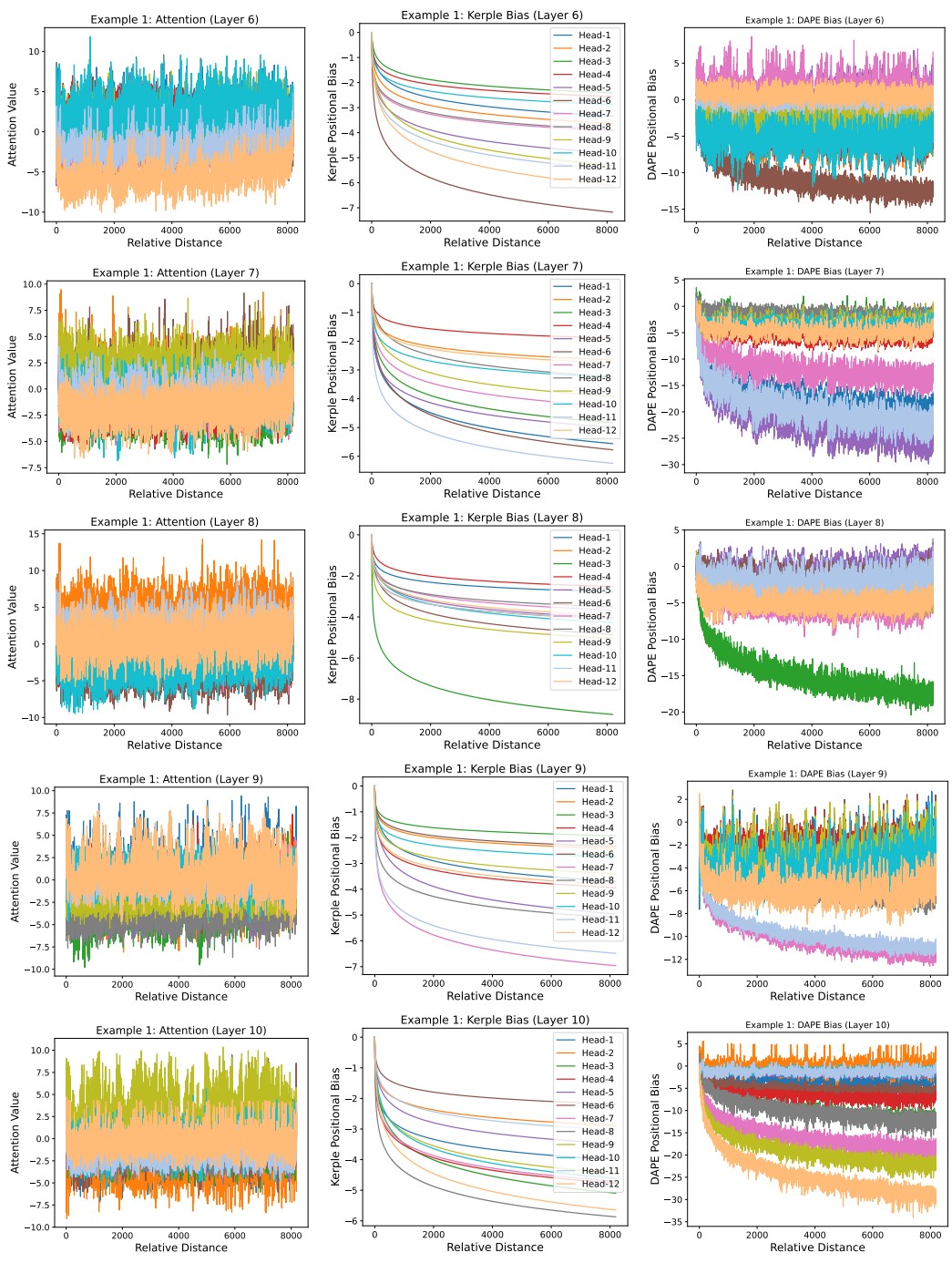

Figure 18: **Evaluation Length 8192 Example 1: Part 2**

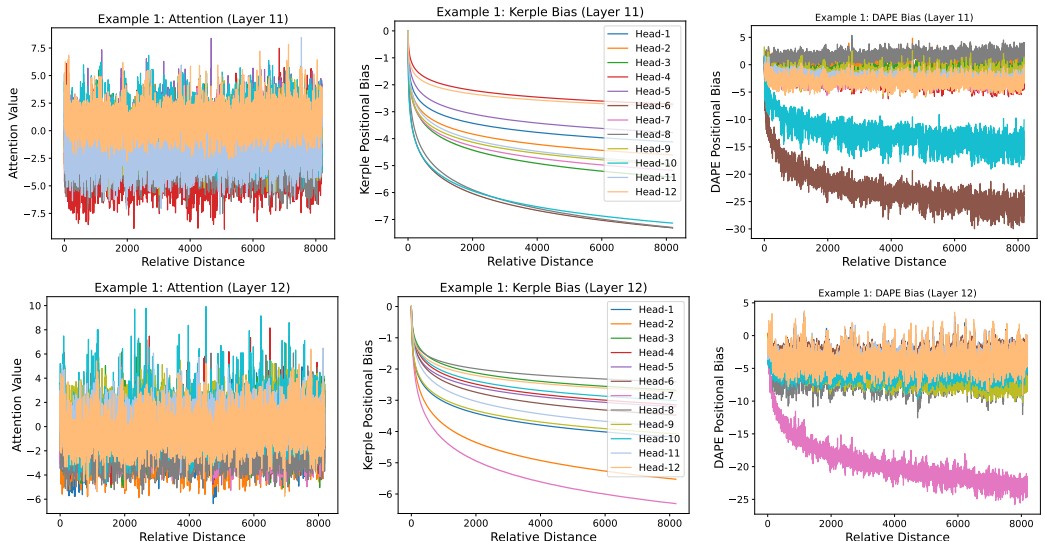

Figure 19: **Evaluation Length 8192 Example 1: Part 3**

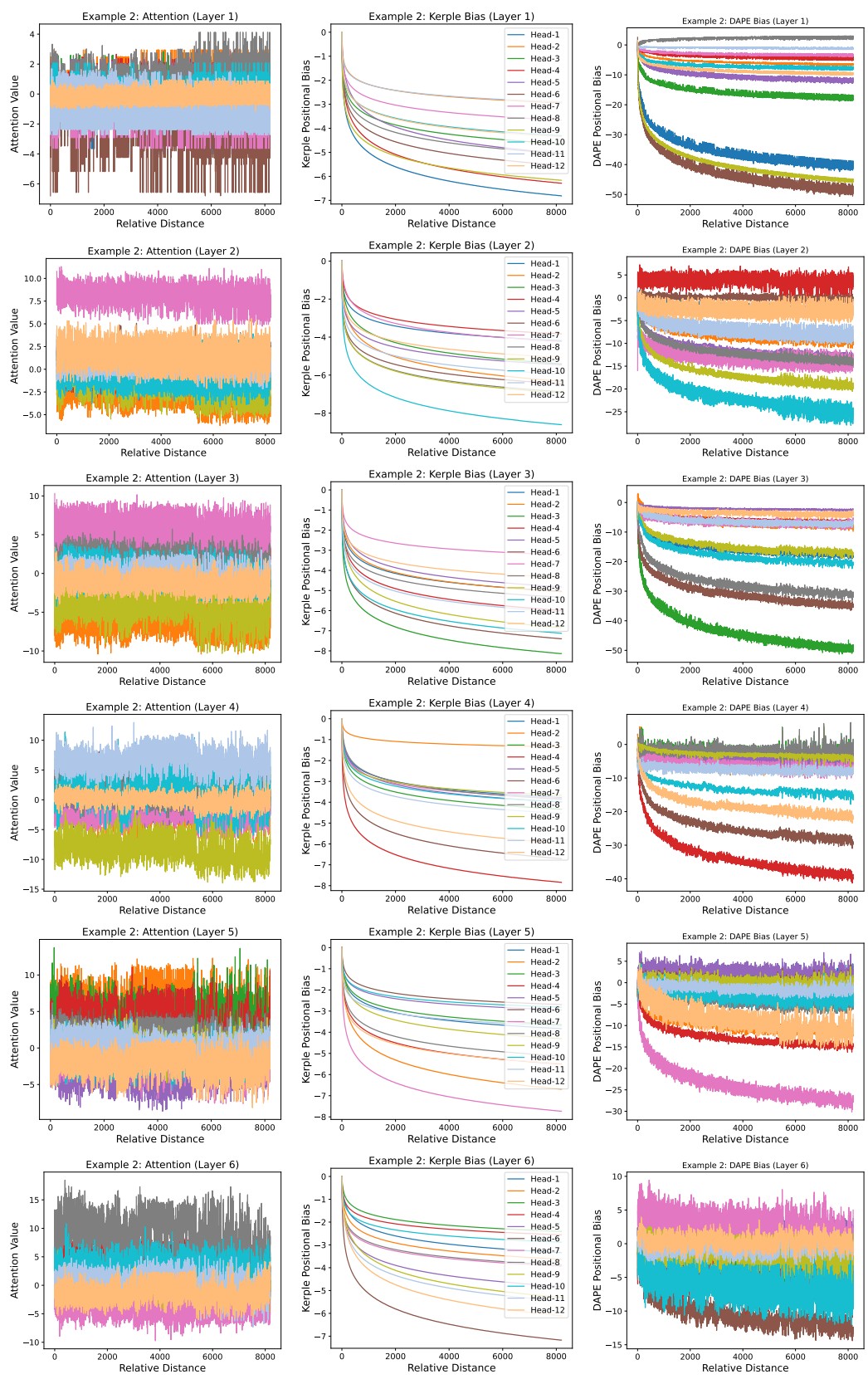

Figure 20: **Evaluation Length 8192 Example 2: Part 1**

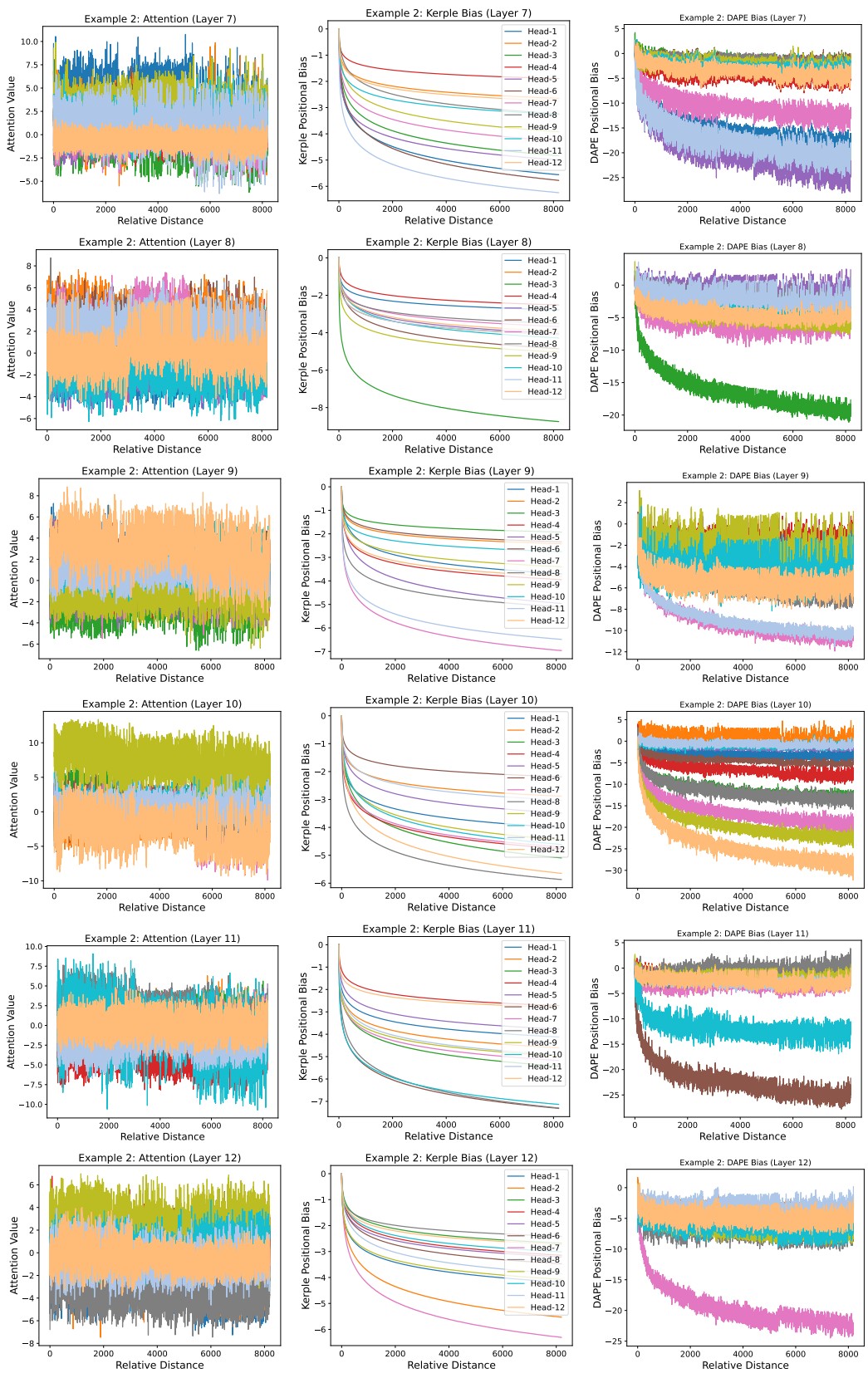

Figure 21: **Evaluation Length 8192 Example 2: Part 2**

# J  Implementation

In this section, we present the implementation of the proposed DAPE module in `PyTorch` [49].

```python
import torch
import torch.nn as nn

class DAPE(nn.Module):
  def __init__(self, head_number=12, mlp_width=32):
    """
    DAPE attention bias module.

    Args:
      num_heads: number of attention heads.
      mlp_width: Width of MLP.
    """
    super(DAPE, self).__init__()

    self.mlp = nn.Sequential(
      nn.Linear(2*head_number, mlp_width),
      nn.LeaklyReLU(),
      nn.Linear(mlp_width, num_heads)
    )

  def forward(self, attention: torch.Tensor, bias: torch.Tensor):
    """
    Args:
      attention: input sequence, which is q^T * k,
          shape [bsz, num_heads, seq_len, seq_len]
      bias: bias matrix, which can be generated by Alibi, Kerple
      FIRE or other additive position encodings
          shape [1, num_heads, seq_len, seq_len]

    Returns:
      attention with DAPE,
      shape [bsz, num_heads, seq_len, seq_len]
    """
    bias_tile = repeat(bias, '1 h T T -> b h T T', b=attention.shape
        [0])

    # Concatenate attention and bias
    attention_bias_concat = torch.cat((attention, bias_tile), dim=1)

    # Rearrange the dimensions for MLP processing
    attention_bias_concat = rearrange(attention_bias_concat, 'b h T
        T -> b T T h')

    # Apply the MLP
    attention_bias_concat = self.mlp(attention_bias_concat)

    # Rearrange back to original dimensions
    attention_bias_concat = rearrange(attention_bias_concat, 'b T T
        h -> b h T T')

    return attention + bias + attention_bias_concat
```

