# OpenReview forum: "DAPE: Data-Adaptive Positional Encoding for Length Extrapolation"
_NeurIPS.cc/2024/Conference — NeurIPS 2024 poster_

### Official Review · Reviewer_QhvW · 2024-07-06

**Soundness:** 3
**Presentation:** 3
**Contribution:** 3
**Rating:** 6
**Confidence:** 5

**Summary:**

This paper proposes a simple learnable positional encoding called CAPE that boosts the length extrapolation performance of Transformer language models.

**Strengths:**

The empirical performance of CAPE is substantially better than previous positional encodings. I also think Figure 1 nicely demonstrates the flexibility of CAPE, i.e., CAPE can learn both local and anti-local attention heads.

**Weaknesses:**

The speed is unbearably slow in my opinion. While I appreciate the honesty of reporting speed differences in Table 1, the authors really need to figure out a way to improve CAPE's training and inference efficiency.

**Questions:**

1. What are the potential ways to improve the speed of CAPE? Please be as concrete as possible.
2. Can the authors test CAPE using needle in a haystack? This way the readers can have a clearer picture of how CAPE is using long context information.

**Limitations:**

None.

---

> ### Author Rebuttal · Authors · 2024-07-31
>
> Dear Reviewer QhvW,
>
> Thank you very much for appreciating our work. We will address your concerns below.
>
> **Q1: The speed of training**
>
> A1: We will answer the question in three parts: 1) **The potential way to improve the speed of CAPE**; 2) **The additional training ratio will gradually decrease with a larger model size**.; 3) **Moreover, CAPE indeed can speed up training**. Please refer to the more detailed answer to **CAPE computation cost** in the **Author Rebuttal by Authors**.
>
> **The potential way to improve the speed of CAPE**:
> * **Reduce the size of $D_{CAPE}$**. The computation cost is $O(hN^2D_{CAPE})$. Therefore, by reducing the $D_{CAPE}$ to be half, the CAPE computation cost will be half.
> * **Algorithm that is more efficient than MLP**. The MLP in CAPE can be changed to other more efficient operations, such as sparse-MLP.
> * **Sparsity**: we can make the MLP operation sparse to speed up CAPE. As shown in Figure 6, our CAPE could work well with only $D_{CAPE}=4$, while the Table 1 count with  $D_{CAPE}=32$
>   * Pruning: use pruning on MLP to remove the redundant parameters.
>   * Dynamic sparsity: we could use dynamic sparse training methods, such as Sparse Momentum, Dynamic Sparse Reparameterization (DSR), Dynamic Sparse Training (DST) and so on.
> * **Data Parallel**: The CAPE consists of fully-connected layer. Therefore, better data parallel can significantly reduce the time.
> * **GPU with high memory width**. The CAPE read/write the matrix with size $N^2$. Therefore, a GPU with a higher memory width will help improve the speed of CAPE.
> * **GPU with high computation compatibility**. The CAPE consists of a multiple-layer perception network, which is computationally dense. Therefore, GPU has better computation compatibility can help speed up the
> * Finally, with the increase in hardware, the cost of CAPE will be acceptable. For example, with the development of GPU, the Large Model gradually is accepted, while it is unimaginable to train a 175B model 10 years ago.
>
> **The additional training ratio will gradually decrease with a larger model size, compared to baseline Kerple.**
>
> The following is the time cost with a training length of 512 with micro_gpu_batch_size 1.
>
> | Method |  350M Total | Ratio(Compared to CAPE-Kerple) | 2.7B Total |Ratio(Compared to CAPE-Kerple) | 6.7B Total | Ratio(Compared to CAPE-Kerple)|
> |------|------|------|------|------|------|------|
> |RoPE|210.01|0.9366| 472.63|1.1187| 635.57|0.8858
> |T5's bias|355.16| 1.5839|537.62|1.2725|808.85|1.1273
> |Alibi|172.60|0.7697|325.95| 0.7715|596.77|0.8317
> |**Kerple**|189.91| **0.8469**|370.32| **0.8765** |661.82|**0.9224**
> |FIRE|248.13|1.1066|432.63| 1.0240|797.68|1.1118
> |**CAPE-Kerple**|224.22|**1.0000** |422.48|**1.0000**|717.46|**1.0000**
>
> Apparently, when the model becomes large, the additional computational cost of CAPE gradually decreases. Therefore, the CAPE may be a potential good choice for an extremely large language model.
>
> **Moreover, CAPE indeed can speed up training, compared to current popular RoPE**
> | Evaluation | RoPE Length 4096 & Batch 1  |Kerple Length 512 & Batch 8|CAPE-Kerple Length 128 & Batch 32 |CAPE-Kerple Length 512 & Batch 8 |CAPE-Kerple Length 1024 & Batch 4 | CAPE-Kerple Length 2048 & Batch 2 | CAPE-Kerple Length 4096 & Batch 1 |
> |------|------|------|------|------|------|------|------|
> |128|38.36|33.04|31.49|32.22|33.22|34.71| 36.65|
> |256|33.21|29.11|28.27|28.32|29.02|30.08|31.57|
> |512|27.33|24.68|24.93|23.88|24.14|24.77|25.68|
> |1024|25.49|23.82|24.31|22.62|22.62|23.09|23.80|
> |2048|23.55|24.03|23.34|21.16|21.00|21.30|21.84|
> |4096|**24.58**|30.76|24.38|**21.79**|21.34|21.45|21.83|
> |8192|152.54|36.81|25.01|21.70|21.12|21.24|21.50
> |Time Cost|**265.48** |117.10 |128.94|**192.45**|314.86|547.78|1217.34
>
> With the same training token, the CAPE with a training length of 512 and batch size of 8 can even with comparable performance with a RoPE training length of 4096 and batch size of 1. Also, the CAPE with a training length of 512 and batch size only takes 192.45ms, while RoPE takes 265.48 ms. Therefore, the CAPE could be a choice for speeding up training in the future.
>
> **Q2: How CAPE is using long context information and haystack test**
>
> A2: We analyze how CAPE uses the long context information via visualization analysis, as shown in Figure 1 and Appendix. According to Figure 1 and Appendix, the CAPE not only helps the model to pay attention to the local information but also helps the model to look at the information far away, with different heads having different functions.
>
> The haystack test, requires more training (including pretrain and alignment) so that the model can follow the instructions to finish the test, while currently, we do not have such resource to train such a model. However, in our CHE benchmark, there is one task named missing duplicate that may be similar to haystack test.
> * haystack test:
>   * It works by embedding specific, targeted information (the “needle”) within a larger, more complex body of text (the “haystack”).
>   * The goal is to assess an LLM’s ability to identify and utilize this specific piece of information amidst a vast amount of data.
> * CHE Benchmark Missing Duplicate Task:
>   * The input is a binary string of the form $ww$ where w is itself a binary
> string. **One token in this string has been hidden, and the network must find out which one it is**,
> by looking at the value at the same position but on the other side of the string. For instance, if
> $x = ab$_$aba$ (i.e., w = aba), then $y = a$.
>   * The goal is also to assess a model's ability to identify and utilize this specific piece of information amidst a vast amount of data.
>  * As shown in Table 2, Kerple only gets 79.06% accuracy, while CAPE-Kerple gets 87.56%. This suggests the ability of CAPE on the haystack test.
>
> If there are any questions, please let us know. And if you think that we have addressed your concerns, could you please consider raising the score? Thank you very much for your support.

---

> > ### Comment · Reviewer_QhvW · 2024-08-10
> > **Thank you for the rebuttal.**
> >
> > I like the additional details provided by the authors. I encourage the authors to continue pushing the efficiency of CAPE. I increased the score to 6.

---

> > > ### Author Response · Authors · 2024-08-10
> > > **Response to Reviewer QhvW**
> > >
> > > Dear Reviewer QhvW,
> > >
> > > Thank you very much for your reply, for improving the score, and for your encouragement. We will continue to refine our work, focusing on both efficiency and effectiveness. We sincerely hope that our efforts will contribute to and inspire the entire research community.

---

### Official Review · Reviewer_NzBk · 2024-07-11

**Soundness:** 2
**Presentation:** 2
**Contribution:** 3
**Rating:** 8
**Confidence:** 3

**Summary:**

Considering the fixed parameters of RoPE may lead to the generalization issue, this paper introduce a dynamic position embedding method named CAPE, where position encoding is depend on the input context. Specifically, CAPE enables testing-time adaptation to input context by using a two-layer LeakyReLU neural network to parameterize the positional bias added to the vanilla attention module. By meticulously selecting the hyper-parameters, CAPE can achieve better performance compared with previous methods.

**Strengths:**

1. The concept is intriguing. Making position embeddings data-independent can potentially enhance the model's performance.
2. The authors conducted numerous experiments on small language models and offered valuable insights.
3. This paper includes several additional experiments shown in the appendix, which provide valuable insights.

**Weaknesses:**

1. CAPE can introduce extra computation, as well as lower the training and inference efficiency
2. The choice of hyperparameters is significant for CAPE, as it is correlated with the model performance and the efficiency.
3. There is no experiments on LLMs.

**Questions:**

1. Can CAPE adapt to LLMs, for example, Llama3-8B, with few post-training steps?
2. What's the context length boundary of CAPE? In this paper, experiments just demonstrate the maximum context length with 8192.
3. The authors claim that "CAPE is semantically dependent and adaptive." What does "semantically dependent" mean? If CAPE is used for a summarization task with different context lengths (perhaps 8192 or 4096), what does "semantic" refer to in the summarization task? Providing a few examples could help clarify the writing.
4. About motivation: Even though positional encoding (PE) is fixed during inference, attention still relies on the input context. What if we consider a scenario where PE simply provides the context index to help the language model distinguish the positions of each token (static), while the attention module dynamically selects key information and provides semantic information to assist the language model? Why does CAPE need to incorporate such a dynamic selection mechanism, which is already learned by LLMs, into the PE?

**Limitations:**

See weakness and question part.

---

> ### Author Rebuttal · Authors · 2024-08-03
>
> Dear Reviewer NzBk,
>
> Thank you very much for appreciating our work. We will address your concerns below.
>
> **Q1: The efficiency of CAPE**
>
> A1: **With the model size increase, the additional computing cost ratio will decrease, compared to baseline Kerple**. Moreover, the CAPE can even speed up the training because a smaller training length of CAPE can still achieve good performance. We discuss this in **CAPE computation cost** in **Author Rebuttal by Authors**.
>
> **Q2: The choice of $D_{CAPE}$**
>
> A2: The CAPE is relatively robust to the choice of $D_{CAPE}$, as shown in Figure 6. To achieve satisfactory performances, we find that CAPE with a modest size (not large $D_{CAPE}$) is sufficient. For convenience, we will copy the results (Arxiv dataset) below.
> |      | 512 | 1024 | 2048 | 4096 | 8192 |
> |------|---------|---------|---------|---------|---------|
> | $D_{CAPE}$ 4 |  4.54   |  4.27   |  4.38   |  4.28   |  4.06   |
> | $D_{CAPE}$ 8 |  4.53   |  4.26   |  4.33   |  4.17   |  3.97   |
> | $D_{CAPE}$ 16|  4.52   |  4.24   |  4.26   |  4.08   |  3.86   |
> | $D_{CAPE}$ 32|  4.50   |  4.22   |  4.22   |  4.04   |  3.82   |
> | $D_{CAPE}$ 64|  4.50   |  4.21   |  4.22   |  4.04   |  3.85   |
> **Q3: The experiment on LLM**
>
> A3: We further conduct experiments on 2.7B and 6.7B model sizes, which proves that the CAPE still works well. We further analyze the result of 2.7B and 6.7B in **Result on Large Model Size 2.7B and 6.7B** in **Author Rebuttal by Authors**.
> We add larger model (2.7 and 6.7BB) experiments in the following, with micr_gpu_batch_size 4 and length 512 (Books Dataset).
> | Model size| Method | 512 | 1024 | 2048 | 4096
> |-------|-------|-------|-------|-------|-------|
> |2.7B|RoPE| 21.01|25.00|48.13|160.59 |
> | | RPE|21.10|21.88|23.59|33.23|
> | | Kerple|21.14|22.08|23.38|27.21|
> | | CAPE-Kerple|20.52|21.01|20.23|19.57|
> |6.7B|RoPE| 20.86|22.27|28.01|110.00
> | | RPE|20.79|21.60|22.32|26.31|
> | | Kerple|20.71|21.57|22.07|24.48|
> | | CAPE-Kerple|20.09|20.54|19.83|19.32|
>
> **Q4: Can CAPE adapt to LLMs, for example, Llama3-8B, with few post-training steps?**
>
> A4: The comment is quite helpful. Although we train from scratch for all transformer models with CAPE (that is the main reason that we did not try very large transformer models due to the computation limitation), it is still possible that we merely train on the CAPE part (i.e., the introduced MLP) but freeze (or fine-tune) all other parameters of transformer models. This strategy will definitely accelerate the training and require fewer post-training steps. We will discuss the possibility of adapting CAPE to LLMs with fewer post-training steps in the paper.
>
> **Q5: What's the context length boundary of CAPE? In this paper, experiments just demonstrate the maximum context length of 8192.**
>
> A5: We further validate CAPE on length 16384, which proves that CAPE still works well. For the length of 32768, the GPU reports out-of-memory. Therefore, we believe that the context boundary of CAPE is more than 16484, while the training length is 128.
> |Method| 128|256|512|1024|2048|4096|8192|16384|
> |-------|-------|-------|-------|-------|-------|-------|-------|-------|
> Kerple|31.96| 29.02| 29.70| 42.74| 56.24| 73.59| 87.03|93.38|
> CAPE-Kerple|31.44| 28.25| 24.93| 24.33| 23.29| 24.32| 24.93|25.33
>
>
> **Q6: What does "semantically dependent" mean?**
>
> A6: Thank you for pointing out. The "semantically dependent" indicates that our position encoding value depends on the semantics (which is the attention score in this paper). For clear presentation, we will change such words to ``context adaptive`` or ``contextually dependent``. And we will also change explain that the context in the paper is attention score.
>
> **Q7: Why does CAPE need to incorporate such a dynamic selection mechanism, which is already learned by LLMs, into the PE**
>
> A7: The reason is that the CAPE has more expressiveness.
>
> Suppose that the ``optimal'' attention mechanism is composed of the key-query multiplication (denoted as $A(x)=XW_Q(XW_K)^T$) and the additive positional encoding bias (denoted as $B(x)$), i.e., $A_{optimal}(x)=XW_Q(XW_K)^T+B(x)$. Here, x and X are the input sequence and the corresponding token embeddings. In previous static PE, $B(x)$ is set as a constant for all input sequence, i.e., $B(x) = B$ and the constant $B$ is optimized across all samples $\{x\}$ during training. However, our main contribution and claim is that the optimal positional encoding should vary for different sequences. Therefore, we proposed the dynamic context-adaptive PE (CAPE), where $B(x)$ depends on both the sequence and the positional information. In contrast with the static PE, the proposed CAPE can adjust dynamically with the input context, and is optimal for each input.
>
> Speaking from higher-level, we see that the general fixed and static PE (even though the PE is learned from data) is an averaged optimal positional encoding over all training samples, while the dynamic PE is context-dependent and is optimal for each sample. That is the core motivation of using dynamic PE rather than the static PE.
>
> If the the optimal solution is $B(x) = B$, the $f(XW_Q(XW_K)^T, B)$ can be reduced to zero as $f(*)$ is a universal approximate function (two-layer mlp with activation). Therefore, our CAPE is at least no worse than the baseline.
>
> If there are any questions, please let us know. And if you think that we have addressed your concerns, could you please consider raising the score? Thank you very much for your support.

---

> > ### Comment · Reviewer_NzBk · 2024-08-11
> > **Response to Authors**
> >
> > I appreciate the authors' response, and I believe conducting experiments on LLMs is essential for this paper. I hope the authors can incorporate the results of these experiments in the revision.
> >
> > However, some questions still confuse me.
> >
> > 1. First, the formulation of A(x) = XW_{Q}(XW_{K})^{T} + B(x). Using RoPE as an example, which can be written as: A(x) = (Q+f(Q))(K + f(K))^{T} = QK^{T} + Qf(K)^{T} + f(Q)K^{T} + f(Q)f(K)^{T}. Is B(x) equal to Qf(K)^{T} + f(Q)K^{T} + f(Q)f(K)^{T} ? or B(x) is equal to f(Q)f(K)^{T}? As the authors mentioned ``B(x) is set as a constant for all input sequence'', however, for RoPE, B(x) has Q and K terms (if B(x) is the former formulation), which is input-depend, rather than a fixed term.
> >
> > 2. Can you explain why CAPE has better extrapolation than other RPE methods, e.g., YaRN, from an Intuitive perspective?
> >
> > 3. Does the ``better extrapolation than other RPE methods'' in the paper mean that CAPE performs better than other RPE methods with the same extrapolation length, e.g., when YaRN and CAPE scale the model context length to both 8192, CAPE has better performance? or does CAPE potentially have a longer context scaling length than YaRN (as you know, YaRN has a limited scaling length)?
> >
> > I hope the authors can respond to those questions.

---

> ### Author Response · Authors · 2024-08-11
> **Response to Reviewer NzBk**
>
> Dear Reviewer NzBk,
>
> Thank you very much for your reply. We will answer your questions below.
>
> **Q1: The RoPE and its B(x)**
>
> A1: **We claim "B(x) is set as a constant for all input sequence" under the discussion of additive RPE**. Usually, the RoPE is not considered as an additive relative position encoding, as discussed in FIRE [3] paper Section 2.2.
>
> We will answer the question about RoPE in two parts: **1) The definition of Additive RPE**; **2) General view of the $B(x)$ and RoPE**
>
> **The definition of Additive RPE, which should have formulation $A(x) = XW_{Q}(XW_{K})^{T} + B$ and B is induced by position encoding function**
>
> According to FIRE paper [3], the additive relative position encoding can be represented by the formulation $A(x) = XW_{Q}(XW_{K})^{T} + B$, while the B $\in R^{N \times N}$ is induced by the **position encoding function** $N^{*2} \to R$. To be more specific, $B=g(D)$, where $g(.)$ is a position encoding function and $D$ is the distance matrix. Usually, the $D$ is the following:
>
> ```python
> D= [0 0 0]
>    [1 0 0]
>    [2 1 0]
> ```
> Therefore, when we are discussing the $A(x) = XW_{Q}(XW_{K})^{T} + B$, usually the $B$ should fulfilled the mentioned requirements. According to FIRE paper [3], the RoPE is not an additive relative position encoding so the RoPE actually does not have such $B$. Therefore, under the original definition, RoPE actually does not have either $B$ or $B(x)$. And for our $B(x)$, we mainly focus how to utilize the additive RPE matrix $B$ and attention score, so that there should be an additive relative position encoding bias matrix $B$.
>
> Therefore, considering the previous definition, we will have the following:
> * **The naive query and key implementation:** $XW_{Q}(XW_{K})^{T}$
> * **The part that utilizes both bias matrix B and $XW_{Q}(XW_{K})^{T}$:** $B(x)$
> * **The part is that fixed after training**: $B$
> * For the previous naive additive position encoding method: $B(x)=B$
> * For our CAPE: $B(x)=B+f(XW_{Q}(XW_{K})^{T}, B)$.
>
> **General view of the $B(x)$ and RoPE**
>
> If we do not consider the definition from FIRE[3] and make the definition of $B$ becomes more general, then we may have the following.
> * **The naive query and key implementation:** $XW_{Q}(XW_{K})^{T}$
> * **The part that removes naive query and key implementation:** $B(x)$
> * **The part is that fixed after training for additive RPE**: $B$
> * For the previous naive additive position encoding method: $B(x)=B$
> * For our CAPE: $B(x)=B+f(XW_{Q}(XW_{K})^{T}, B)$.
> * For RoPE: $B(x)=Qf(K)^{T} + f(Q)K^{T} + f(Q)f(K)^{T}$. From the general additive RPE perspective, this may be why RoPE is better than its baseline Sinusoidal encodings.
>
> **Q2: Why CAPE is better than other RPE methods e.g.**
>
> A2: Compared to other RPEs e.g. YaRN (YaRN is also a great work), we further improve the performance of the current great method Kerple and FIRE[3] via dynamic position encoding to get better performance. Also, YaRN improves performance via sequence-length-related adjustment.
>
> * **Our baseline is powerful**. We may check the FIRE paper[3] Figure 1 (on page 2). Figure 1 of the FIRE paper compares the performance between Kerple, and FIRE and other RPEs (including YaRN). The results prove that Kerple and FIRE achieve relatively good performance.
> * **CAPE solves the baseline Kerple and FIRE's limitation, whose position encoding is fixed after training.** As we claim in our paper, though additive RPE achieves good performance, its position encoding is fixed after training. Hence, we propose to further improve the additive relative position encoding performance by dynamically adjusting the position encoding via attention score.
> * Therefore, our method could achieve relatively better performance than other methods.
>
> **Q3: The definition of "better extrapolation than other RPE methods"**.
>
> A3: The better extrapolation means: with the same experiment setting (training length, training tokens, and so on), our proposed method could achieve better performance on evaluation length $T_{eval}$, while $T_{eval}$ is larger than training length $T_{train}$.
>
> For example, we train CAPE-Kerple with **the maximum training length of 1024**, and we could say that CAPE-Kerple could have better extrapolation performance if CAPE-Kerple achieves better performance on evaluation length 8192. We follow the definition of length extrapolation from previous works [1][2][3].
>
> **Thank you very much for your reply. If there is any other question, please let us know.**
>
> Reference:
>
> [1] Press, O., Smith, N., & Lewis, M. Train Short, Test Long: Attention with Linear Biases Enables Input Length Extrapolation. ICLR, 2021.
>
> [2] Chi, T. C., Fan, T. H., Ramadge, P. J., & Rudnicky, A. (2022). Kerple: Kernelized relative positional embedding for length extrapolation. NIPS, 2022
>
> [3] Li, S., You, C., Guruganesh, G., Ainslie, J., Ontanon, S., Zaheer, M., ... & Bhojanapalli, S. Functional Interpolation for Relative Positions improves Long Context Transformers. ICLR, 2024.

---

> ### Comment · Reviewer_NzBk · 2024-08-11
> **Re-Response to Authors**
>
> Ok, I appreciate the authors' professional response.
>
> In your paper, on line 233, the authors mention, "As shown in Figure 4 and Table 5, CAPE consistently outperforms established baselines such as RoPE" (I know you mention Figure 2, please correct this in the final revision). It is surprising for me to know that Additive Position Embedding works better than RoPE-based methods.
>
> I also read the general response by the authors, the ``Q2: Result on Large Model Size 2.7B and 6.7B (Reviewer tTna, Reviewer NzBk)'' also surprised me, as it indicates that additive RPE can achieve much better results than RoPE under the context scaling settings.
>
>
> 1. Can you explain why the additive RPE is better than the RoPE-based model here intuitively?
>
> 2. Are all the experiments in Figure 2 fair and consistent? Including model size, training data, etc.
>
> 3. From the results in Figure 2, it seems that the mainstream RoPE is not as effective as Additive RPE. What do you think about this phenomenon? Should the mainstream RPE now use additive RPE rather than cumulative RPE like RoPE?
>
> I hope the authors answer the above questions, which are very important for my judgment of your work.

---

> > ### Author Response · Authors · 2024-08-11
> > **Response to Reviewer NzBk (Part 1/2)**
> >
> > Dear Reviewer NzBk,
> >
> > Thank you very much for your notice in Line 233, we have revised it for the final revision.
> >
> > And thank you very much for your question. We will answer your question below.
> >
> > **Q1: Can you explain why the additive RPE is better than the RoPE-based model here intuitively**
> >
> > A1: The additive RPE has two attributes: 1) could have explicitly long-term decay (local position pattern); 2) could have the anti-local pattern. We will explain it below step by step.
> >
> > **Explicitly long-term decay (local position pattern)**
> > * The RoPE paper claims that long-term decay is important for long context, and the RoPE achieves it via implicitly long-term decay.
> > * As the additive RPE has the formulation $A(x)=XW_Q(XW_K)^T+B$, we could implement the explicitly long-term decay (local position pattern) pattern via a bias matrix that B has a negative value.
> >    * For example,  $B(i,j) = -r_1\log(1+r_2|i-j|)$ (logarithmic variant) , where $r_1, r_2>0$ are learnable scalars.
> >
> > **The anti-local pattern: emphasize far away keys more**
> > * For long-context, we cannot abandon long-distance information, otherwise it will become local attention. Therefore, the anti-local position pattern is important so that the model could pay attention to long-distance information.
> > * The RoPE has long-term decay so that with distance increase, the long-distance information weight will be smaller. **Therefore, RoPE does not have anti-local position pattern.**
> > * The additive RPE FIRE successfully achieves anti-local pattern, as shown in its paper Figure 4.
> > * Our CAPE-Kerple also successfully achieves anti-local position pattern, as shown in Figure 1 and Appendix F. For example, with the distance increases, the bias value is non-decreasing.
> >
> > Therefore, as the additive RPE could have both local and anti-local position patterns, it may be better.
> >
> > **Q2: Are all the experiments in Figure 2 fair and consistent? Including model size, training data, etc.**
> >
> > A2: **We promise that the experiments are fair and consistent**. We follow the training protocol from Kerple. **And we only change the position encoding method for different experiments.** The model size of Figure 2 is 125M, and the model size of Figure 4 is 350M. The experiment setting is shown in Appendix B. And we directly copy it here.
> >
> > |                             |             | **125M**    |             | **350M**   |
> > |-----------------------------|-------------|-------------|-------------|------------|
> > | Training sequence length    |             | 512         |             | 512        |
> > | Batch size                  |             | 32 × 8      |             | 32 × 8     |
> > | Number of iterations        |             | 50k         |             | 50k        |
> > | Dropout prob.               |             | 0.0         |             | 0.0        |
> > | Attention dropout prob.     |             | 0.0         |             | 0.0        |
> > | Attention head              |             | 12          |             | 16         |
> > | Feature dimension           |             | 768         |             | 1024       |
> > | Layer number                |             | 12          |             | 24         |
> > | Optimizer                   |             | Adam        |             | Adam       |
> > | Optimizer parameter betas   |             | [0.9, 0.95] |             | [0.9, 0.95]|
> > | Learning rate               |             | 6e-4        |             | 3e-4       |
> > | Precision                   |             | float16     |             | float16    |

---

> ### Author Response · Authors · 2024-08-11
> **Response to Reviewer NzBk (Part 2/2)**
>
> **Q3: From the results in Figure 2, it seems that the mainstream RoPE is not as effective as Additive RPE. What do you think about this phenomenon?**
>
> A3: The following is our personal opinion. The mainstream method is RoPE because of three reasons: 1) The timing of RoPE and additive RPE ; 2) Within training length, RoPE has comparable performance with additive RPE; 3) The development of LLaMA.
>
> **The timing of RoPE and additive RPE**
> The RoPE paper is on arxiv from 20 April 2021, while there was no Alibi (27 Aug, 2021), Kerple (20 May, 2022), or FIRE (3 Oct, 2023).
>
> **Within training length, RoPE has comparable performance with additive RPE.**
> As shown in Figure 2 and Appendix 5, for the performance within training length, the proposed RoPE achieves close or similar performance with Kerple and FIRE. For example, with a training length 512, RoPE achieves 4.5755 ppl, Kerple achieves 4.5817 and FIRE achieves  4.5741. Therefore, for the performance within training length, the different is not very large. The additive RPE presents its superiority for length extrapolation, and the length extrapolation problem has become important recently.
>
> Therefore, as previously we mainly considered performance within training length, the RoPE is enough.
>
> **The development of LLaMA**
> * LLaMA uses RoPE. Therefore, if anyone is working on open-source LLM, then they will use LLaMA so that they will use RoPE.
> * Therefore, we could find that the YaRN, CLEX, ChunkLLaMA or other works that focus on length extrapolation or long-context all focus on RoPE.
> * Also, currently data is relatively important so the architecture from LLaMA to LLaMA 3 is not changed a lot. Hence, the current mainstream position encoding method is RoPE.
>
> **Q4: Should the mainstream RPE now use additive RPE rather than cumulative RPE like RoPE?**
>
> A4: It is an interesting question, and we are also curious about it. Both additive PE and RoPE-based PE try to incorporate the key-query similarity with the positional information of tokens. However, they adopt different operations, using addition and multiplication operations respectively. In this paper, we developed an additive context-adaptive PE. It is hard to say which kind of PE (additive or RoPE-based) will be the mainstream finally. [RoPE is widely recognized and used in Llamma models.]
>
> Based on the current evidence and the experiment results, additive RPE may be a better choice.
> * The FIRE has prove the it could achieve better performance than RoPE, whatever within training length or beyond training length.
> * CAPE further helps achieve better performance within training. As proved in our paper (Figure 2 and Table 5), the CAPE could help additive RPE achieves better performance within training length.
> * CAPE further helps achieve better length extrapolation performance. Also, as proved in our paper, the CAPE could help additive RPE achieve better performance beyond training length.
> * Therefore, based on the current evidence and experiment results, the additive RPE may be better.
>
> **Thank you very much for your constructive comments. If there is any further question, please let us know.**

---

> > ### Comment · Reviewer_NzBk · 2024-08-12
> > **Final Response to Authors**
> >
> > Dear Authors
> >
> > Glad to hear the authors' constructive responses, both the general and individual responses.
> >
> > The authors conduct numerous experiments during the discussion period and offer many insightful opinions and explanations about the work.
> >
> > I hope this paper can be accepted by NIPs, as it may contribute to the LLM community.
> >
> > BTW, I hope the authors make sure the final revision includes some modifications during the discussion phase, as it can better help readers understand the work and drive progress in the field of long context models.
> >
> > I have no further questions and will raise my score from 6 to 8.
> >
> > Best,
> >
> > Reviewer NzBK

---

> > > ### Author Response · Authors · 2024-08-12
> > > **Response to Reviewer NzBk**
> > >
> > > Dear Reviewer NzBk,
> > >
> > > Thank you very much for your reply, for your encouragement and support. **We promise that our final revision will include some modifications during the discussion phase**, including but not limited to the experiment of large model size, the context length boundary of CAPE, the modification of "semantically dependent", the discussion of RoPE and so on.
> > >
> > > Again, thank you very much for your attention and support to this work, and wish you a good day.

---

### Official Review · Reviewer_CfZF · 2024-07-12

**Soundness:** 3
**Presentation:** 3
**Contribution:** 3
**Rating:** 7
**Confidence:** 3

**Summary:**

The paper proposes context-adaptive positional encoding. The paper proposes a 2-layered MLP to non-linearly integrate positional bias information (can be computed using prior methods like Alibi, FIRE, and Kerple) and query-key content-based dot product values representing semantic relations across different heads to dynamically create a contextual position-informed matrix that can additively modulate the self-attention matrix.

**Strengths:**

1. The idea is reasonably motivated.
2. The empirical results are promising for length extrapolation in language modeling. The experiments on Chomsky's hierarchy tasks are a nice touch.

**Weaknesses:**

1. If I understand correctly, the original relative encoding (from Shaw et al. and then the one in Transformer XL) could also count as contextual. There, the position-based additive values are computed based on the dot product of queries and distance encodings (treated as keys). So, the context can influence the distance-related bias through the query representations. This could be better discussed in the paper. I would also be curious how Transformer XL style relative encoding would perform in the language modeling datasets.

2. There is some hit in the computation expense. Although it is manageable, the authors don't treat each head independently but use the concatenation of heads with a shared hidden state.

**Questions:**

Minor suggestions:

* It Would be good to refer to the appendix implementation section G when discussing function f and the multi-head CAPE in the main paper.
* Perhaps it is better to present the pseudocode with einops-style syntax.

**Limitations:**

There isn't an explicit section purely for limitations, but there is a section for computational cost analysis, which may illuminate some limitations (and authors treat the section like a limitation section in the checklist). Overall, it's mostly adequate.

---

> ### Author Rebuttal · Authors · 2024-08-02
>
> Dear Reviewer CfZF,
>
> Thank you very much for appreciating our work. We will address your concerns below.
>
> **Q1: The performance of transformer-xl relative encoding**
>
> A1: We have shown the performance of transformer-xl below, which also presents great length extrapolation performance. Also, the Relative in Table 2 is just the transformer-xl  relative encoding. The experiment is conduct on 125M model with training length 512 and batch size 32.
> | Method     | 512 | 1024 | 2048 | 4096 | 8192 |
> |------------|-----------|-----------|-----------|-----------|-----------|
> | Rope       | 19.75     | 261.39    | 411.24    | 635.80    | 762.86    |
> | T5's bias  | 19.67     | 19.45     | 33.41     | 141.94    | 347.36    |
> |Transformer-XL|19.40|19.23|19.17|21.21|23.23|
> | Alibi      | 20.04     | 19.75     | 20.17     | 20.50     | 21.31     |
> | Kerple     | 19.83     | 19.20     | 20.49     | 28.33     | 40.95     |
> | FIRE       | 19.77     | 21.09     | 103.14    | 308.58    | 484.55    |
> | Cape-Kerple| **19.25**     | **18.28**     | **17.20**     | **17.58**     | **17.85**     |
>
> According to the experiment, the transformer-xl achieves good performance on length extrapolation. Therefore, this also suggests that the position encoding should interact with attention/query/key to further improve the performance.
>
> **Q2: The computation expense**
>
> A2: Thank you very much for your precious comment on the compuation cost. Yes, we concatenate the attention and bias on the head dimension, and then we use an mlp to process them to dynamically adjust the position encoding values. We have further analyze the CAPE cost in **CAPE computation cost** in **Author Rebuttal by Authors**.
>
> **Q3: Mention appendix implementation section G when discussing function f and the multi-head CAPE in the main paper.**
>
> A3: Thank you very much for your suggestion. We will revise the paper with the following sentence:
>
> **Original:** It then outputs $h$-dimensional vectors, where each element corresponds to the CAPE for the respective head.
>
> **Revised:**   It then outputs $h$-dimensional vectors, where each element corresponds to the CAPE for the respective head. We have shown the code implementation in Appendix G.
>
> **Q4: Present the pseudocode with einops-style syntax**
>
> A4: Thank you very much for your suggestion. We will add the einops-style syntax in Appendix G.
> ```python
> import torch
> import torch.nn as nn
> from einops import rearrange, repeat
> class CAPE(nn.Module):
>     def __init__(self, head_number=12, mlp_width=12):
>         """
>         CAPE attention bias module.
>
>         Args:
>           head_number: number of attention heads.
>           mlp_width: Width of MLP.
>         """
>         super(CAPE, self).__init__()
>
>         self.mlp = nn.Sequential(
>             nn.Linear(2 * head_number, mlp_width),
>             nn.LeakyReLU(),
>             nn.Linear(mlp_width, head_number)
>         )
>
>     def forward(self, attention: torch.Tensor, bias: torch.Tensor):
>         """
>         Args:
>           attention: input sequence, which is q^T * k,
>              shape [bsz, num_heads, seq_len, seq_len]
>           bias: bias matrix, which can be generated by Alibi, Kerple
>           FIRE or other additive position encodings
>              shape [1, num_heads, seq_len, seq_len]
>
>         Returns:
>           attention with CAPE,
>           shape [bsz, num_heads, seq_len, seq_len]
>         """
>         # Repeat the bias for batch size
>         bias_tile = repeat(bias, '1 h l1 l2 -> b h l1 l2', b=attention.shape[0])
>
>         # Concatenate attention and bias
>         attention_bias_concat = torch.cat((attention, bias_tile), dim=1)
>
>         # Rearrange the dimensions for MLP processing
>         attention_bias_concat = rearrange(attention_bias_concat, 'b h l1 l2 -> b l1 l2 h')
>
>         # Apply the MLP
>         attention_bias_concat = self.mlp(attention_bias_concat)
>
>         # Rearrange back to original dimensions
>         attention_bias_concat = rearrange(attention_bias_concat, 'b l1 l2 h -> b h l1 l2')
>
>         return attention + bias + attention_bias_concat
> ```
>
> If there are any questions, please let us know. And if you think that we have addressed your concerns, could you please consider raising the score? Thank you very much for your support.

---

> > ### Comment · Reviewer_CfZF · 2024-08-08
> > **Rebuttal Response**
> >
> > Thank you for the rebuttal. It generally addresses my initial concerns. The new experiments should improve the paper. I would encourage adding more discussion in comparison with Transformer XL as well. I also do not see any technical issue unlike tTna and your rebuttal makes sense to me. I maintain my acceptance score for now.
> >
> > But two questions:
> >
> > 1. I noticed in your bigger LLM experiments that you provided in the response that AliBi is missing. Is there a reason for that? Do you have any comments on it?
> >
> > 2. I may have missed it the first time around - but how does it stack up against something like xPos? Can you comment more on that method. I noticed that the xPos paper is in the references (42), but I couldn't find the context where it is cited (if at all).

---

> ### Author Response · Authors · 2024-08-09
> **Author Response**
>
> Dear Reviewer CfZF,
>
> Thank you very much for appreciating our work. We will follow your suggestion to add the new experiment results  and discuss the transformer-xl in our paper, and we will update it immediately when we are allowed to revise the paper. Also, we will answer the two questions mentioned in the following:
>
> **Q1: The Experiment of Alibi**
>
> A1: The experiment of Alibi is the following.
>
> **For model size 2.7B:**
> | Model size| Method | 512 | 1024 | 2048 | 4096
> |-------|-------|-------|-------|-------|-------|
> |2.7B|RoPE| 21.01|25.00|48.13|160.59 |
> | | Alibi|21.23|22.17|22.91|23.22|
> | | T5's bias (RPE)|21.10|21.88|23.59|33.23|
> | | Kerple|21.14|22.08|23.38|27.21|
> | | CAPE-Kerple|20.52|21.01|20.23|19.57|
>
> **For model size 6.7B**:
> | Model size| Method | 512 | 1024 | 2048 |
> |-------|-------|-------|-------|-------|
> |6.7B|RoPE| 20.86|22.27|28.01|
> | | Alibi|20.79|21.63|22.45|23.22|
> | | T5's bias (RPE)|20.79|21.60|22.32|
> | | Kerple|20.71|21.57|22.07|
> | | CAPE-Kerple|20.09|20.54|19.83|
>
>
> The reason for missing Alibi experiment result: with the model size increase, we face some engineering challenges:
> * Our implementation is based on the framework GPT-NeoX, which already implements the Alibi position encodings.
> * You may notice that our maximum evaluation length is 8192 for 125M and 350M, while the maximum evaluation length for 2.7B is 4096 and the maximum evaluation length for 6.7B is 2048. The reason is that the Alibi position encoding faces out-of-memory challenges for the evaluation length 8192 for the 2.7B model size and 4096 for the 6.7B model size.
> * Therefore, considering two aspects ( 1. evaluation as long as possible; 2. the most popular position encoding method is RoPE), we mainly present the results of RoPE, T5's bias, Kerple and CAPE-Kerple in rebuttal so that the evaluation length can be longer (evaluation length 4096) for both 2.7B model size and 6.7B model size.
> * If you would like to see more experiment results, please let us know. We will try our best to finish the experiment as soon as possible.
>
> **Q2: Discussion with XPos**
>
> A2: The CAPE is designed for Additive Relative Position Encoding, while XPos is an improved version of RoPE (Note: RoPE is Relative Position Encoding, but NOT Additive Relative Position Encoding). Therefore, CAPE and XPos focus on different directions of position encodings.
>
>
> **The Difference between XPos and CAPE**
>
> The XPos:
>    * The implementation of RoPE: $f_q(q,n)=qe^{i\theta n}$
>    * The implementation of XPos: $f_q(q,n)=qe^{\xi n+i\theta n}$
>    * Apparently, the XPos is an improved version of RoPE, while XPos degrades to RoPE when $\xi$ becomes zero.
>
> The CAPE:
>    * The implementation of Additive Relative Position Encoding: $A(x)=XW_Q(XW_K)^T+B$. The $B$ is the bias matrix, which could be Alibi, Kerple, FIRE, or other potential additive relative position encoding methods.
>    * The implementation of CAPE: $A(x)=XW_Q(XW_K)^T+B + f(XW_Q(XW_K)^T, B)$.
>    * Apparently, the CAPE can be applied to any potential additive relative position encoding methods.
>
> **The performance of XPos and CAPE-Kerple**
>
> The previous paper BiPE [1] conducted experiments using XPos on the Arxiv dataset. As shown in the BiPE [1] paper's Figure 4, with the training length 1024, the perplexity of XPos increases quickly from about 6 ppl (at evaluation length 1024) to about 16 ppl (at evaluation length 6144), while our proposed method (CAPE-Kerple) decreases the ppl from 5.21 (evaluation length 1024) to 5.00 ppl (at evaluation length 8192) with training length 128. This suggests that our method should be better than the mentioned XPos.
>
>
> Therefore, the XPOS is an improved version of RoPE (note that RoPE is not additive relative position encoding), while CAPE generally improves the performance of the additive relative position encodings, including Alibi, Kerple, FIRE and so on.
>
> **Finally, if you have any questions or would like to discuss anything, please let us know. We will try our best to share our opinions or conduct experiments.**
>
> Reference:
>
> [1] He, Z., Feng, G., Luo, S., Yang, K., He, D., Xu, J., ... & Wang, L. (2024). Two Stones Hit One Bird: Bilevel Positional Encoding for Better Length Extrapolation. arXiv preprint arXiv:2401.16421.

---

> > ### Comment · Reviewer_CfZF · 2024-08-10
> >
> > Thank you for the additional details. This could be included in the paper/appendix.

---

> ### Author Response · Authors · 2024-08-10
> **Response to Reviewer CfZF**
>
> Dear Reviewer CfZF,
>
> Thank you very much for your reply, and thank you very much for your support. We promise that we will include the additional details/experiments in the paper/appendix.

---

### Official Review · Reviewer_tTna · 2024-07-15

**Soundness:** 2
**Presentation:** 3
**Contribution:** 2
**Rating:** 3
**Confidence:** 5

**Summary:**

This paper offers a doable solution for long context tasks. The paper introduces the Context-Adaptive Positional Encoding (CAPE) method to enhance transformer model adaptability and flexibility in processing long input lengths and contexts. To overcome the limitations of static positional encodings such as Absolute Positional Encoding and Relative Positional Encoding, the proposed method adjusts the positional encodings based on input context and learned fixed priors. Experimental evaluation on Arxiv, Books3, and CHE shows that the proposed method significantly improves model performance in length generalization.

**Strengths:**

1. The long-context ability of LLMs is crucial and fundamental for advancements in the LLM domain, significantly impacting downstream tasks.
2. The proposed method is intuitive and straightforward to implement and understand.
3. The performance of the proposed method, as indicated by PPL, appears promising and is well-demonstrated in Figures 2, 3, and 4.

**Weaknesses:**

1. The major concern is in the experimental evaluation part. The experiment is limited to PPL, which is insufficient for verifying long-context ability.
2. The experiments are limited to a very small LLM model - a 124M transformer, which is not enough to verify the proposed method, as large LLMs and small LLMs behave quite differently.
3. The proposed method is not convincing to me. The proposed method is mainly in Equations (2) and (3). In Equation (2), the proposed method does not seem promising. Considering the next step ( $h(*)$ ) in the neural network, it will be $ h(A_{\text{CAPE}}(X)) = h(XW_Q(XW_K)^\top + f(XW_Q(XW_K)^\top, B)) = g(XW_Q(XW_K)^\top, B) $. Could the author please elaborate on this?
4. Similarly, for Equation (3), $ A_{\text{CAPE}}(X) = XW_Q(XW_K)^\top + B + f(XW_Q(XW_K)^\top, B) $, it has a similar problem to Equation (2).
Due to the major weakness in the experiment and the unreasonable proposal of the bias term, I would like to reject this paper.

**Questions:**

See Weaknesses

**Limitations:**

See Weaknesses

---

> ### Author Rebuttal · Authors · 2024-07-31
>
> Dear Reviewer tTna,
>
> Thank you for the detailed review. We will address your concerns below.
>
> **Q1: The major concern is in the experimental evaluation part.**.
>
> A1: We have presented the experimental results besides PPL, as shown in Page 9 Section 4.7 and Appendix D Experiments on Chomsky Hierarchy Evaluation Benchmark (the benchmark and its variants are used in previous works), which **uses accuracy as evaluation metrics.**
>
> Four convenience, we directly copy the result of CHE Benchmark here. We train with 200K steps and length 40, while we test on length 500.
> The random accuracy is 50%, except for modular arithmetic simple, cycle navigation, bucket sort, solve equation, and modular arithmetic brackets, where it is 20%.
> $\dagger$$\dagger$$\dagger$ denotes permutation-invariant tasks, which are expected to be solved without positional information.
>
> | Level | Task | Learned | sin/cos | RoPE | Relative | ALiBi | Kerple | FIRE | | Alibi | Kerple | FIRE |
> |-------|------|---------|---------|------|----------|-------|--------|------|-------|--------|-------|------|
> |       |      | **Randomized**  | **Randomized**  | **Randomized**  | **Randomized**  | **Randomized**  | **Randomized**  | **Randomized**  | | **CAPE (Ours)**  | **CAPE (Ours)**  | **CAPE (Ours)**  |
> | R     | even pairs | 50.04   | 91.27 | 99.98 | 96.60   | 73.52  | 57.50  | 73.86 | | 99.99 | 99.58  | **100** |
> |       | modular arithmetic simple | 19.95   | 20.39 | 21.35 | 20.84   | 20.02  | 21.79  | 21.09 | | 23.58 | **24.47** | 24.46 |
> |       | parity check$\dagger$$\dagger$$\dagger$ | 50.14   | 50.52 | 50.05 | 50.09   | 50.09  | 50.07  | **50.97** | | 50.30 | 50.07  | 50.04 |
> |       | cycle navigation$\dagger$$\dagger$$\dagger$ | 24.97   | 25.37 | 27.63 | 26.95   | 24.64  | 29.47  | 28.41 | | 22.99 | **34.53** | 27.54 |
> | DCF   | stack manipulation | 59.92   | 65.92 | 61.49 | 64.73   | 66.42  | 66.93  | 69.33 | | 68.18 | **72.04** | 70.90 |
> |       | reverse string | 52.76   | 67.28 | 65.23 | 65.59   | 71.09  | 71.54  | 65.89 | | 73.37 | 70.74  | **76.40** |
> |       | modular arithmetic brackets | 31.00   | 30.70 | 31.25 | 31.74   | 30.56  | 24.79  | 30.92 | | 31.34 | **32.37** | 31.50 |
> |       | solve equation | 20.00   | 19.97 | 21.85 | **22.93** | 19.92  | 21.15  | 22.06 | | 20.03 | 22.49  | 22.42 |
> | CS    | duplicate string | 52.77   | 65.44 | 64.97 | 67.66   | 65.13  | 66.72  | 69.03 | | 70.84 | **72.95** | 72.71 |
> |       | missing duplicate | 50.38   | 49.78 | 63.37 | 72.34   | 74.21  | 79.06  | 79.27 | | 83.41 | 87.57  | **89.17** |
> |       | odds first | 52.77   | 58.61 | 61.00 | 61.57   | 59.88  | 62.59  | 63.28 | | 63.78 | **67.08** | 66.34 |
> |       | binary addition | 54.63   | 55.78 | 55.59 | 56.96   | 54.72  | 56.35  | 55.70 | | 59.71 | **60.88** | 56.62 |
> |       | compute sqrt | 50.47   | 51.11 | 51.88 | 51.63   | 50.63  | 51.11  | 50.80 | | 51.64 | 51.33  | **52.46** |
> |       | bucket sort$\dagger$$\dagger$$\dagger$ | 98.32   | 98.92 | 98.12 | 99.31   | 98.45  | 99.38  | **99.57** | | 99.38 | 98.81  | 99.37 |
>
>
> **Comparative performance improvements.**
> CAPE consistently enhanced performance across various tasks, especially on permutation-variant tasks. Specifically, CAPE improved upon Alibi and FIRE's results in all 11 tested permutation-invariant tasks. Similarly, it outperformed Kerple in 10 of these tasks.
>
> **Q2: Experiments on large model size**
>
> A2: The following is the result of the experiment on 2.7B and 6.7B, with training length 512 and micro_gpu_batch_size 4. We further discuss the experiment on large model size in **Result on Large Model Size 2.7B and 6.7B** in **Author Rebuttal by Authors**
>
> | Model size| Method | 512 | 1024 | 2048 | 4096
> |-------|-------|-------|-------|-------|-------|
> |2.7B|RoPE| 21.01|25.00|48.13|160.59 |
> | | RPE|21.10|21.88|23.59|33.23|
> | | Kerple|21.14|22.08|23.38|27.21|
> | | CAPE-Kerple|20.52|21.01|20.23|19.57|
> |6.7B|RoPE| 20.86|22.27|28.01|110.00
> | | RPE|20.79|21.60|22.32|26.31|
> | | Kerple|20.71|21.57|22.07|24.48|
> | | CAPE-Kerple|20.09|20.54|19.83|19.32|
>
> **Q3: Explanation of Equation 2 and Equation 3**
>
> A3: The next step of our operation is $softmax$. Therefore, $h(A_{CAPE}(x))=softmax(XW_Q(XW_K)^T+f(XW_Q(XW_K)^T, B))!=softmax(XW_Q(XW_K)^T+B)=h(A(x))$, while $A(x)$ is the naive transformer attention score calculation without CAPE.
>
> Let us explain the attention module in Transformer step by step:
> * Step 1 (Calculate attention score via query $XW_Q$ and $XW_K$):
>     *  Previous implementation of $A_{score}=XW_Q(XW_K)^T+B$
>     *  CAPE implementation of     $A_{score}=XW_Q(XW_K)^T+B+f(XW_Q(XW_K)^T, B)$
> * Step 2 (use softmax on attention-score, row-wise. Therefore, the **next step is** $\textbf{softmax}$):
>     * $A_{scoreSoftmax}=softmax(A_{score})$
> * Step 3 $A_{scoreSoftmax}$ and value $XW_K$ to get embedding of each token:
>     * output=$A_{scoreSoftmax}XW_V $
>
> If we understand correctly, the neural network h(∗) mentioned in the comment corresponds to the feedforward network layer (FFN), that follows the attention layer. We would like to clarify that the MLPs introduced in the CAPE model are not duplicated. In CAPE, these MLPs dynamically adjust the positional encodings based on context information (**CAPE: Step 1**). Subsequently, softmax operations are applied across attention score, row-wise. It is important to note that the CAPE's adjustments, applied during the attention phase, do not directly alter the token values but act on the attention computation. However, the FFN layer that follows attention modifies (**FFN: after Step 3**) each token through a nonlinear transformation.
>
> If there are any questions, please let us know. And if you think that we have addressed your concerns, could you please consider raising the score? Thank you very much for your support.

---

> ### Author Response · Authors · 2024-08-13
> **Kindly Remind of Discussion Period**
>
> Dear Reviewer tTna,
>
> We would like to thank you again for your detailed reviews. We have updated the experiment results and the explanation of our method in the above response.
>
> As the discussion period will be closed soon, we would appreciate it if you could let us know if our responses have addressed your concerns and whether you still have any other questions. We would be happy to do any follow-up discussion or address any additional comments.
>
> Again, thank you very much for your attention to our work.

---

### Author Rebuttal · Authors · 2024-08-01

Dear all reviewers:

We sincerely appreciate the reviewers for the time and efforts on the review. We first address some common questions, followed by detailed responses to each reviewer separately. We hope our responses clarify existing doubts. We will really appreciate it if Reviewer tTna can kindly reconsider the decision, provided that the main comments are well addressed.

**Q1: CAPE computation cost (Reviewer CfZF, Reviewer NzBk, Reviewer QhvW)**

**The additional training ratio will gradually decrease with a larger model size, compared to baseline Kerple.**.

Theorically Analysis:
 * The cost of Feed-Forward Network is: $O(Nd_{head}^2d_{hidden}^2)$=$aNd_{head}^2d_{hidden}^2$, where a is a constant, N is the sequence length, $d_{head}$ is the attention head number and $d_{hidden}$ is the dimension for attention calculation.
* The cost of Attention: $O(N^2d_{head}d_{hidden})$=$bN^2d_{head}d_{hidden}$, where b is a constant.
*  The additional cost of CAPE: $O(N^2d_{head}d_{cape})$=$cN^2d_{head}d_{cape}$, where c is a constant.
* The cost ratio is $\frac{aNd_{head}^2d_{hidden}^2+bN^2d_{head}d_{hidden}}{aNd_{head}^2d_{hidden}^2+bN^2d_{head}d_{hidden}+cN^2d_{head}d_{cape}}$=$\frac{ad_{head}d_{hidden}^2+bNd_{hidden}}{ad_{head}d_{hidden}^2+bNd_{hidden}+cNd_{cape}}$. Therefore, with the fixed sequence length and $d_{CAPE}$, with the model becomes larger (with bigger $d_{head}$ and $d_{hidden}$), the additional cost ratio of CAPE will greatly become smaller. Also, we have shown in Figure 6 that $d_{cape}$ still works well with very small value, such as 4.


The following is the time cost with a training length of 512 with micro_gpu_batch_size 1 on Books3 dataset.

| Method |  350M Total | Ratio(Compared to CAPE-Kerple) | 2.7B Total |Ratio(Compared to CAPE-Kerple) | 6.7B Total | Ratio(Compared to CAPE-Kerple)|
|------|------|------|------|------|------|------|
|RoPE|210.01|0.9366| 472.63|1.1187| 635.57|0.8858
|T5's bias|355.16| 1.5839|537.62|1.2725|808.85|1.1273
|Alibi|172.60|0.7697|325.95| 0.7715|596.77|0.8317
|**Kerple**|189.91| **0.8469**|370.32| **0.8765** |661.82|**0.9224**
|FIRE|248.13|1.1066|432.63| 1.0240|797.68|1.1118
|**CAPE-Kerple**|224.22|**1.0000** |422.48|**1.0000**|717.46|**1.0000**

Apparently, when the model becomes large, the additional computational cost of CAPE gradually decreases. Therefore, the CAPE may be a potential good choice for an extremely large language model.

**Moreover, CAPE indeed can speed up training, Compared to current popular RoPE**
| Evaluation | RoPE Length 4096 & Batch 1  |Kerple Length 512 & Batch 8|CAPE-Kerple Length 128 & Batch 32 |CAPE-Kerple Length 512 & Batch 8 |CAPE-Kerple Length 1024 & Batch 4 | CAPE-Kerple Length 2048 & Batch 2 | CAPE-Kerple Length 4096 & Batch 1 |
|------|------|------|------|------|------|------|------|
|128|38.36|33.04|31.49|32.22|33.22|34.71| 36.65|
|256|33.21|29.11|28.27|28.32|29.02|30.08|31.57|
|512|27.33|24.68|24.93|23.88|24.14|24.77|25.68|
|1024|25.49|23.82|24.31|22.62|22.62|23.09|23.80|
|2048|23.55|24.03|23.34|21.16|21.00|21.30|21.84|
|4096|**24.58**|30.76|24.38|**21.79**|21.34|21.45|21.83|
|8192|152.54|36.81|25.01|21.70|21.12|21.24|21.50
|Time Cost|**265.48** |117.10 |128.94|**192.45**|314.86|547.78|1217.34

With the same training token, the CAPE with a training length of 512 and batch size of 8 can even with comparable performance with a RoPE training length of 4096 and batch size of 1. Also, the CAPE with a training length of 512 and batch size only takes 192.45ms, while RoPE takes 265.48 ms. Therefore, the CAPE could be a choice for speeding up training in the future.

Finally, with developments in hardware, the cost of CAPE will become more manageable. For instance, the development of GPUs has led to the widespread acceptance of large models, something that would have been unimaginable 10 years ago when training a 175-billion parameter model.

**Q2: Result on Large Model Size 2.7B and 6.7B (Reviewer tTna, Reviewer NzBk)**

A2:  We add larger model (2.7 and 6.7BB) experiments in the following, with micr_gpu_batch_size 4 and length 512 on Books3 dataset.
| Model size| Method | 512 | 1024 | 2048 | 4096
|-------|-------|-------|-------|-------|-------|
|2.7B|RoPE| 21.01|25.00|48.13|160.59 |
| | RPE|21.10|21.88|23.59|33.23|
| | Kerple|21.14|22.08|23.38|27.21|
| | CAPE-Kerple|20.52|21.01|20.23|19.57|
|6.7B|RoPE| 20.86|22.27|28.01|110.00
| | RPE|20.79|21.60|22.32|26.31|
| | Kerple|20.71|21.57|22.07|24.48|
| | CAPE-Kerple|20.09|20.54|19.83|19.32|

According to the result, we can find that the proposed CAPE still works well, whatever the model size is 2.7B or 6.7B. With 2.7B model size, RoPE achieves 21.01 on evaluation length 512 and 160.50 on evaluation 4096, while our CAPE-Kerple achieves 20.52 and 19.57 respectively. Also, CAPE-Kerple achieves the best performance, whatever the model size is 2.7B or 6.7B from evaluation length 512 to 8192. This suggests that our proposed CAPE has great scalability.

---

### Comment · Area_Chair_a2or · 2024-08-11
**Question about learnt additional bias**

Dear Authors,

Thanks for all your responses and extra experiments. I have additional question related to your discussion with reviewers.

It was shown that ALiBi is in the nutshell window attention (https://arxiv.org/pdf/2212.10356). Also there was a discussion in the KERPLE about similar behaviour as well log version of KERPLE is not window attention and far away positions are actually used and the longer context improves overall model performance.

Now, regarding your method:
- As in Fig.1 you showed that the learned bias is both local and anti-local - what is the intuition how the model solves the generalization problem of not seen positions? As anyway you will have $B$ values which are not seen during training, so we could expect that $f$ will generalize well to the test-time $B$.
- Could you clarify why for CAPE in contrast to all other baselines you observe improvement with the longer context, while other degrade always? This seems to be a new phenomenon - so you are showing that you are not only generalizing but also the longer context (maybe) is used effectively. Could it be that with even longer context the trend will continue? Any intuition from the learned representation why this happens? Maybe compare Fig 1 for position 2k, 4k, 8k to see the trend of change in how the bias is learnt?
- Why do you need still to have $B + f(QK^T, B)$? Why not just $f(QK^T, B)$?

AC.

---

> ### Comment · Reviewer_NzBk · 2024-08-11
> **Have a similar question with AC**
>
> Dear Authors,
>
> Similar to question#2 raised by AC, I also have this question (see Question #1 and #3 posed in Re-Response to Authors by NzBk).
>
> The authors can combine these questions to respond, thx.
>
> Reviewer NzBk

---

> > ### Author Response · Authors · 2024-08-11
> > **Response to Area Chair a2or and Reviewer NzBk (Part 1/3)**
> >
> > Dear Area Chair a2or and Reviewer NzBk,
> >
> > Thank you very much for your question. We will address them below.
> >
> > **Q1: Nutshell window attention**
> >
> > A1: We have discussed the question in our paper Section 4.1 Comparisons with Baselines and Table 2 (you can find the Table 2 in the rebuttal to Reviewer tTna). To prove that our model could sufficiently use the entire sentence information, we conducted the experiments in Table 2.
> >
> > **CAPE enhances intra-length performance, indicating that its lower perplexity may come
> > from thorough utilization of entire sentences but not disregarding long-distance information
> > (Also proved in Figure 1).** Compared to Alibi, Kerple, and FIRE, the adapted versions CAPE-Alibi, CAPE-Kerple, and CAPE-FIRE demonstrate better intra-length
> > performance. With the growing sequence length, the Alibi tends to transition from full attention
> > to almost local attention, and this is why Alibi is worse than most baselines within training length
> > but better beyond training lengths. The results (in Table 5) indicate that the superior intra-length performance of CAPE is statistically significant, with a p-value less than 0.05. Therefore, the
> > consistent intra-length performances across various training lengths indicate that the lower perplexity
> > of CAPE results from effectively utilizing the entire sequence, rather than focusing on local parts and
> > neglecting long-distance information.
> >
> > **Experiment on tasks that need the entire sequence information (as shown in Table 2)**. As the some methods could cheat perplexity (such as Alibi), we further conduct experiments on benchmarks that need to sufficiently utilize the whole sentence. The results are shown in Table 2 Page 9.
> > * **CAPE works better on permutation-variant tasks.** CAPE (with Kerple and FIRE) presented the
> > best performance in 10 out of 11 permutation-variant tasks (which require positional information),
> > achieving the second-best performance in the SOLVE EQUATION task. This underscores the efficacy
> > of CAPE with semantic adaptivity in handling permutation-variant challenges.
> > * **CAPE’s performance on permutation-invariant tasks.** In tasks that are permutation-invariant,
> > where positional information is non-critical, CAPE demonstrated comparable performance. Notably,
> > CAPE-Alibi achieved scores of 50.30 on PARITY CHECK and 99.38 on BUCKET SORT tasks, compared to the highest scores of 50.97 and 99.57, respectively, demonstrating competitive performances.
> >
> >
> > **Q2: what is the intuition how the model solves the generalization problem of not seen positions?**
> >
> > A2: **Recent works prove that the transformer can be generalized to unseen potions**, such as Kerple and FIRE. CAPE helps to enhance such ability via dynamically adjusting position encodings for different input sequences, and its performance is actually highly related to the baseline method.
> > | Method        | 128   | 256   | 512   | 1024  | 2048  | 4096  | 8192  | 16384 |
> > |---------------|-------|-------|-------|-------|-------|-------|-------|-------|
> > | Kerple        | 31.96 | 29.02 | 29.70 | 42.74 | 56.24 | 73.59 | 87.03 | 93.38 |
> > | CAPE-Kerple   | 31.44 | 28.25 | 24.93 | 24.33 | 23.29 | 24.32 | 24.93 | 25.33 |
> >
> > **The CAPE-Kerple achieves 24.33 ppl on evaluation length 1024 and 25.33 on evaluation length 16384. Therefore, the performance of CAPE-Kerple will also degrade as the performance of Kerple becomes worse.**
> >
> > * As the Area Chair a2or mentioned above, the Kerple uses the log to reduce the distribution gap between seen position id and unseen position id.
> > * The recent work FIRE even uses the formulation $B(i,j)=f(\frac{i-j}{\max(\{L, i\})})$ to reduce the gap between seen and unseen position id. Whatever how long the context is, the $\frac{i-j}{\max(\{L, i\})} \in [0,1]$ so that the gap between training and validation becomes even smaller.
> > * There are also other papers that suggest transformers can generalize to unseen positions, such as "Transformers Can Achieve Length Generalization But Not Robustly".
> > * Therefore, based on the previous work, the bias matrix $B$ already could partly solve the unseen position id problem.
> > * Hence, the one potential problem becomes: how can we further better utilize the bias matrix $B$ to get better position encodings?
> >    * One potential solution: we try to design a better position encoding function $N^{*2}\to R$ to get a better bias matrix. There are related works, including Alibi, Kerple and FIRE.
> >    * Another potential solution is: all we need is a better attention score and position encodings so why not dynamically adjust the bias matrix $B$ to get better position encoding? Following the direction, we developed the CAPE and it actually works well, without additional operations.
> > * Intuitively, our CAPE may help the additive RPE to be learnable and optimized, as the $B+f(QK^T,B)$ could degrade to $B$ if $f(QK^T,B)$ is not necessary. Also, our CAPE is context-adaptive position encoding so that CAPE could dynamically adjust position encoding for different sequences.

---

> ### Author Response · Authors · 2024-08-11
> **Response to Area Chair a2or and Reviewer NzBk (Part 2/3)**
>
> **Q3: Could you clarify why for CAPE in contrast to all other baselines you observe improvement with the longer context, while other degrade always?**
>
> A3: The CAPE performance is highly related to the baseline method, such as Kerple or FIRE. **Though CAPE could usually improve additive RPE method performance, CAPE may achieve worse performance if the baseline additive RPE method collapses.**
>
> The following are the results of FIRE and CAPE-FIRE on the Books3 dataset. With a training length of 512, CAPE can consistently improve performance.
>
> | Method (train 512)             | 512 | 1024 | 2048 | 4096 | 8192  |
> |------------------------|-------------|-------------|-------------|--------------|--------------|
> |RoPE|19.62|170.93|336.76|426.50|559.13|
> | Kerple             | 19.68       | 19.07       | 20.44       | 28.34        | 39.32        |
> | FIRE               | 19.67       | 19.52       | 34.96       | 129.33       | 278.78       |
> | CAPE-Kerple (Ours) | 19.23       | 18.23       | 17.15       | 17.63        | 17.88        |
> | CAPE-FIRE (Ours)   | 19.14       | 18.39       | 18.22       | 21.45        | 28.86        |
>
> And if we change the training length from 512 to 128, we will have the following results.
> | Method (Train 128)           | 128     | 256     | 512     | 1024     | 2048      | 4086       |8192       |
> |-------------------|-------|-------|-------|--------|--------|--------|---------|
> |RoPE|32.08 | 29.80 | 33.12 | 102.51 | 234.86 | 383.83 | 513.31 |
> | Kerple            | 31.96 | 29.03 | 29.71 | 42.75  | 56.25  | 73.59  | 87.03   |
> | CAPE-Kerple (Ours)| 31.45 | 28.25 | 24.93 | 24.34  | 23.30  | 24.33  | 24.93   |
> | **FIRE**              | **31.91** | **29.06** | **32.61** | **110.29** | **321.53** | **534.54** | **802.76**  |
> | **CAPE-FIRE (Ours)**  | **31.42** | **28.40** | **26.81** | **34.03**  | **80.98**  | **333.44** | **1107.74** |
>
> According to the experiments, we can see that the CAPE-FIRE performance could be even worse when the FIRE performance almost collapsed (which is 802.76 ppl at evaluation 8192).
>
> Therefore, the CAPE performance is highly related to the baseline additive RPE performance. CAPE could improve performance when the baseline method could achieve acceptable performance under the current situation, while the performance may degrade if the baseline method collapses under the current situation.
>
> **Q4: Could it be that with even longer context the trend will continue?**
>
> A4: As we discussed above, the performance of CAPE is highly related to the baseline method performance, such as Kerple and FIRE. For Kerple, our method still works well when the training length 128 and the evaluation length is 16384.
> | Method        | 128   | 256   | 512   | 1024  | 2048  | 4096  | 8192  | 16384 |
> |---------------|-------|-------|-------|-------|-------|-------|-------|-------|
> | Kerple        | 31.96 | 29.02 | 29.70 | 42.74 | 56.24 | 73.59 | 87.03 | 93.38 |
> | CAPE-Kerple   | 31.44 | 28.25 | 24.93 | 24.33 | 23.29 | 24.32 | 24.93 | 25.33 |
>
>
> **Q5: Any intuition from the learned representation why this happens? Maybe compare Fig 1 for position 2k, 4k, 8k to see the trend of change in how the bias is learnt?**
>
> A5: We have presented more examples of the Figure 1 visualization in Appendix F, with length 512, 2048 and 8192. And the training length 512. **Intuitively, our CAPE works better because it has both local and anti-local position patterns, which is achieved by different heads.**
> * **The CAPE could really adjust the bias matrix value for different samples**. In Appendix F, for each validation length (512, 2048 or 8196), we present the visualization of two different samples. Let us first focus on the evaluation length 512.
>    * For sample 1 with evaluation length 512, its layer 2's $4^th$ head output bias value is mostly smaller than 5.
>    * For sample 2 with evaluation length 512, its layer 2's $4^th$ head output bias value could be larger than 5.
>    * This suggests that the CAPE can really adjust the bias matrix for different samples.
> * **Different heads have different functions (trends), whatever the evaluation length is 512, 2048, or 8192 .**
>    * Whatever the evaluation length 512, 2048 or 8192, it seems that the different layers' different heads have their own functions after training, which may help improve the model performance.
>    * For the original attention score, it seems that different heads may not have different functions. This may be why NoPE does not work very well.
>    * For the Kerple bias, all heads pay attention to local information.
>    * For our CAPE, the different heads have different functions
>       * The layer 1's head 8 is used to pay attention to long-distance information.
>       *  The layer12's head 7 is used for the local information.

---

> > ### Author Response · Authors · 2024-08-11
> > **Response to Area Chair a2or and Reviewer NzBk (Part 3/3)**
> >
> > **Q6: Why do you need still to have $B+f(QK^T,B)$ and why not just $f(QK^T,B)$**
> >
> > A6: We thank for pointing out this. We have discussed performance of $B+f(QK^T,B)$ and $f(QK^T,B)$ in Section 4.3 Different Variants of CAPE Figure 5.
> >
> > In equation (2) at the methodology part of the paper, we introduce the context-adaptive PE (denoted as f(*)) as a surrogate to the previous fixed PE. The original formulation of CAPE is attention score plus the context-adaptive PE f(*). As the residual connection is a popular technique in deep learning, we further introduce a variant of CAPE with the residual connection as shown in equation (3). Experimental results (shown in Section 4.3, Figure 5) on both two formulations of CAPE indicate that they have very minor performance differences.
> >
> > Specifically, at a training length of 128, $B+f(QK^T,B)$ records a score of 5.00 and  $f(QK^T,B)$
> > scores 5.03 at an evaluation length of 8192.
> > With a training length of 512, $B+f(QK^T,B)$ achieves a score of 3.70, and $f(QK^T,B)$ scores 3.69 at
> >  an evaluation length of 8192. We use $B+f(QK^T,B)$ just to hope that it will be easier for $B+f(QK^T,B)$ to degrade to $B$ if needed, to be sure that CAPE is better than baselines.
> >
> > **Thank you very much for your constructive comments. If there is any further question, please let us know.**

---

> > > ### Comment · Area_Chair_a2or · 2024-08-12
> > > **Reply**
> > >
> > > Dear Authors,
> > >
> > > Thanks for clarifications and pointing out to the plots I missed in the Appendix.
> > >
> > > Some further thoughts:
> > > - Figure 5 actually has different variants of the residual but it doesn't have version where you don't have explicit residual on $B$. By any chance did you run configuration where there is only $f$ w/o residual on $B$? Wanna to exclude inductive bias on $B$ here.
> > > - Looking at the Appendix F if I correctly interpret the leftmost plot - this is the final attention. So, from the rightmost plots it seems that some heads learn $f$ to be uniform across position (so attend to all despite the position index) and some learn kind of windowing (with log decay, or some other dependency) similar to KERPLE. But interesting that resulting attention for all heads looks in average per position (if I do smoothing of the left column plots across x-axis) as uniform. Having this, I wonder if you run standard baseline on the noPos which was shown and proved before can learn positional embedding in decoder only models because of the causal masking.
> > >
> > > AC.

---

> ### Author Response · Authors · 2024-08-12
> **Response to Area Chair a2or**
>
> Dear Area Chair a2or,
>
> Thank you very much for your reply. We will answer your questions below.
>
> **Q1: Wanna to exclude inductive bias on $B$ here**
>
> A1: We present the CAPE $QK^T+f(QK^T, B)$ in the following tables (Figure 5).
>
> |   Arxiv Dataset (Train Length 128)         | 128       | 256       | 512       | 1024       | 2048       | 4096       | 8192       |
> |------------|---------|---------|---------|---------|---------|---------|---------|
> | $QK^T+B$ (baseline)          | 8.28  | 7.09  | 6.29  | 9.93  | 16.97 | 26.14 | 31.91 |
> | $QK^T+B+f(QK^T+ B)$       | 8.21 | 6.99 | 5.41 | 5.27 | 5.46 | 5.40 | 5.17 |
> | $QK^T+f(QK^T, B)$ (**w/o inductive bias $B$**)         | 8.18 | 6.95 | 5.35 | 5.18 | 5.34 | 5.25 | **5.03** |
> | $QK^T+B+f(QK^T, B)$       | 8.20 | 6.98 | 5.39 | 5.21 | 5.36 | 5.25 | **5.00** |
>
>
> |   Arxiv Dataset (Train Length 512)         | 512   | 1024  | 2048  | 4096  | 8192  |
> |------------|-------|-------|-------|-------|-------|
> | $QK^T+B$  (baseline) |4.56  | 4.38  | 5.58  | 8.13  | 10.91  |
> | $QK^T+B+f(QK^T+ B)$  | 4.50  | 4.21  | 4.19  | 3.98  | 3.75   |
> | $QK^T+f(QK^T, B)$ (**w/o inductive bias $B$**)      | 4.47  | 4.18  | 4.14  | 3.94  | **3.69**   |
> | $QK^T+B+f(QK^T, B)$     | 4.50  | 4.20  | 4.18  | 3.95  | **3.70**   |
>
> According to the question, **the CAPE variant $QK^T+f(QK^T, B)$ may be what we need**. If we need to try other variants, please let us know.
>
>
> **Q2: Looking at the Appendix F if I correctly interpret the leftmost plot - this is the final attention.**
>
> A2: The leftmost plot is $QK^T$. We have described the figure Attention, Figure Kerple Bias and CAPE bias in Figure 1. We directly copy it here.
> * (1) The Attention (the leftmost plot) is $QK^T$, which is the naive attention implementation
> * (2) The Kerple bias (the middle plot) is $B$;
> * (3) The CAPE (with Kerple) bias (rightmost plot) is $f(QK^T,B)$.
> * **Therefore, the final attention is (1)+(2)+(3). We promise that we will provide the final attention visualization in the final revision.**
>
> **Q3: Baseline of NoPE**
>
> A3: We have run the baseline of NoPE, as shown in Figure 1. For convenience, we directly copy our results here.
> |   Arxiv Dataset (Train Length 128)         | 128      | 256      | 512        | 1024        | 2048        | 4096        | 8192        |
> |------------|--------|--------|----------|----------|----------|----------|----------|
> | Nope       | 8.42   | **8.64**   | 122.04   | 549.48   | 880.23   | 1264.52  | 1366.45  |
> | RoPE       | 8.29   | **8.55**   | 22.79    | 69.68    | 142.95   | 249.74   | 274.15   |
> | T5's bias  | 8.29   | 7.11   | 5.84     | 6.42     | 8.70     | 15.54    | 33.30    |
>
> |    Arxiv Dataset (Train Length 512)        | 512        | 1024        | 2048        | 4096        | 8192        |
> |------------|----------|----------|----------|----------|----------|
> | Nope       | 4.68   | **31.79**  | 1867.46  | 4666.60  | 5334.85  |
> | Rope       | 4.57   | **43.62**  | 144.05   | 278.87   | 297.06   |
> | T5's bias  | 4.54   | 4.51   | 13.87    | 121.45   | 338.06   |
>
> **With sufficient training, the NoPE may achieve relatively good performance**
>    * When the training length is 128, the NoPE is worse than RoPE, from the evaluation length 128 to 8192
>    * When the training length is 512, the NoPE could get better performance than RoPE on evaluation length 1024.
>    * Therefore, NoPE actually could potentially have acceptable length extrapolation performance, while it needs sufficient training.
>
> **For the NoPE attention visualization**
>
> According to your suggestion, we also visualize NoPE.
>
> **On Arxiv Dataset**
> * With evaluation length 512:
>    * **For the early layer (layer 1 to 2)**: its attention distribution is similar to some leftmost plots, where all heads look in average per position. This may because early layer does not receive sufficient implicit position information.
>    * **For the middle layer (layer 3 to 7)**
>       *  Most attention heads present long-term decay. This decay may follow either a linear relationship with distance or a logarithmic pattern.
>    * **For the latter layer (layer 8 to 12)**
>       * The attention distribution gradually becomes relatively similar to the early layer.
>
> * With the increase in the evaluation length, such long-term decay becomes more obvious.
>
>
> **On Books3 Dataset**
> * The Books3's attention distribution in different layers is relatively similar to the Arxiv dataset's head pattern.
> * Also, we observed the non-local attention head pattern, such as layer 2's head 9. For example, with the distance increase, the attention score of layer 2's head 9 does not decrease.
> * This may explain why additive RPE and CAPE are better: explicitly help provide the needed local and anti-local attention pattern.
>
> Finally, **we promise that we will provide the NoPE attention visualization and discussion in the final revision.**
>
> **Thank you very much for your constructive comments. If there is any further question, please let us know.**

---

> > ### Comment · Area_Chair_a2or · 2024-08-12
> > **Reply**
> >
> > Dear authors,
> >
> > Thanks for additional experiments and clarifications on those I missed from some Tables.
> >
> > Just to clarify, I am trying to understand if we see something entirely new compared to all prior positional embeddings with the new idea of learning the positional embedding and inductive bias you brought.
> >
> > Regarding the total attention: if I do in mind sum over all 3 plots, this will lead still to the kind of KERPLE style final attention. What happens is adjusting the proper decay of positions, showing that maybe log heuristic is not ideal, and something better is needed. The latter is what CAPE is doing exactly. Also I would then disagree that CAPE learns anti-local attn: on its own it does it, but then the total bias is $B + f(...)$ and there is no anti-local attn for it (I do (2) + (3)). In my view right now, it is a bit overstated in the paper, as CAPE is about finding better functions of position decay compared to KERPLE e.g. (which is totally fine, but there is no new principled pattern in CAPE in general). And then yep, it is very dependent on the base method, as $f$ is trying to correct a bit the slope of decay, but cannot change it drastically for the unseen positions.
> >
> > Regarding NoPos - I don't understand why you don't have perfect generalization to the longer sequences - I expect here performance being worse overall than other positional embeddings e.g. for the length same as training, but then it should be uniform and independent from the sequence length as you don't have issues on the position extrapolation at all. Your numbers are not uniform, but they are even significantly worse for generalization to longer than training sequences, which is strange to me. Any explanation / intuition?
> >
> > AC.

---

> ### Author Response · Authors · 2024-08-12
> **Resubmit below**
>
> After the reply submission, it seems that the openreview system does not send email and notification. Therefore, we resubmit the reply below

---

> ### Author Response · Authors · 2024-08-13
> **Response to Area Chair a2or**
>
> Dear Area Chair a2or,
>
> Thank you very much for your reply. We will answer your question below.
>
> **Q1: The attention pattern of Kerple and CAPE-Kerple**
>
> A1: To purely analyze the function $f$, we further analyze the visualization with  $QK^T+f(QK^T,B)$ (The Equation 2 for CAPE in our paper) so that the total bias matrix is $f(QK^T,B)$, without being affected by the residual $B$.
>
> **For CAPE-Kerple bias with CAPE   formulation $QK^T+f(QK^T,B)$**.
>
> According to the visualization, the total bias matrix $f(QK^T,B)$ presents clearly anti-local attention pattern. For example, the layer 1's head 8 and head 2 present clearly anti-local attention pattern, which is relatively similar to the anti-local pattern shown in the Appendix F layer 1's $f(QK^T,B)$ in formulation $QK^T+B+f(QK^T,B)$. Hence, we could have confidence to say that the proposed CAPE bias could present anti-local attention patterns. The function $f$ not only changes the slope of decay but also provides the anti-local attention pattern if needed.
>
> **The final attention pattern comparison between Kerple and CAPE-Kerple**
>
> We also further visualize the Kerple final attention(which is $QK^T+B$) and CAPE-Kerple final attention (which is $QK^T+f(QK^T,B)$) . As the CAPE total bias matrix $f(QK^T,B)$ could clearly provide the anti-local attention pattern, the CAPE-Kerple final attention ($QK^T+f(QK^T,B)$)  successfully clearly presents the anti-local attention pattern on some attention heads so that Kerple final attention pattern and CAPE final attention pattern are different.
>
> **We promise that we will provide the visualization of Kerple and CAPE-Kerple ($QK^T+f(QK^T,B)$) in the final revision**
>
>
>
>
>
> **Q2: Why you don't have perfect generalization to the longer sequences? Any explanation/intuition?**
>
> A2: **The previous work already proves that the NoPE performance may not be uniform and independent from the sequence length, such as NoPE Paper[1] and FIRE [2].**
>
> * According to the NoPE paper[1], the performance of NoPE is not uniform and independent of the sequence length. For example, for the NoPE Paper [1] Figure 3, the performance of NoPE will be worse if the evaluation length is larger than the training length. The NoPE paper claims that there is potential for NoPE to have better extrapolation performance than other position encodings, but it does not claim that the NoPE performance is uniform and independent from the sequence length.
> * The FIRE paper also compares the performance of NoPE in FIRE paper's Figure 1. According to the figure, we can find that the NoPE performance becomes worse (perplexity also increases) when the evaluation length is larger than the training length.
> * Our personal opinion.
>    * As claimed by NoPE Paper [1], the NoPE implicitly learns the position encoding information, which may not be easy to learn and is inefficient.
>    * Therefore, we may need to sufficiently train the model if we use NoPE.
> * **Therefore, according to the previous works (such as [1][2]), the NoPE performance may not be uniform and independent of the sequence length.**
>
>
> **Thank you very much for your constructive comments. If there is any further question, please let us know.**
>
>
> Reference:
>
> [1] Kazemnejad, A., Padhi, I., Natesan Ramamurthy, K., Das, P., & Reddy, S. (2024). The impact of positional encoding on length generalization in transformers. Advances in Neural Information Processing Systems, 36.
>
> [2] Li, S., You, C., Guruganesh, G., Ainslie, J., Ontanon, S., Zaheer, M., ... & Bhojanapalli, S. Functional Interpolation for Relative Positions improves Long Context Transformers. In The Twelfth International Conference on Learning Representations.

---

> > ### Comment · Area_Chair_a2or · 2024-08-14
> > **Reply**
> >
> > Dear Authors,
> >
> > Thanks again for involving in the discussion.
> >
> > I have several more questions regarding overall numbers you shared:
> > - FIRE proposed a variant of mapping positions into (0, 1) so the problem then becomes interpolation rather than extrapolation and they show in their paper that the method extrapolates to unseen positions better than others. Why then in your comparison in Figures 2, 3, 4 and above Tables FIRE is the worst in the generalization? Even worse than RoPE? This on its own seems to be strange.
> > - Why did you decide to select 128 context lengths for training? It seems unreasonable, and thus results on generalization you present may not be aligned with prior results as e.g. in FIRE 2048 context length is used, and AliBi e.g. 512 and 1024.
> > - Could you explain the choice of Arxiv and Books3 data as e.g. prior works used WikiText, C4, OpenWebText2, GitHub?
> > - In KERPLE paper authors show that being trained on 512 context they are able to extrapolate with even lowering perplexity on Arxiv till 16k context (see in their paper Table 3 and 5), while in your results above KERPLE degrade in performance very quickly, e.g. it doesn't generalize already at 2048 length from Fig 2, while in KERPLE paper Table 3 it shows improvements over increasing the context length at test time?
> > - Looking at Fig. 3 it seems that extrapolation depends a lot on the training context size and larger size used, better we extrapolate to longer sequences (maybe not surprising), but then it is not clear how all results on 128 length are helpful for overall conclusions and claims of the paper.
> > - Figures 2, 3 and 4 demonstrate that all prior papers which claimed extrapolation at least till 16k do not hold and only AliBi is able to generalize to that length. This I found contradicting all prior works tbh. But then in Figure 5 you are showing that actually AliBi also does not extrapolate, as baseline red there is AliBi and at 2048 length it degrades significantly.
> > - For Nopos: I agree it is not uniform, but what I meant - it is kept uniform for some time and then degrades, outperforming others in the generalization. Your results with Nopos are questionable, as the perplexity degrades super quickly to the wildly off values. This is inconsistent with finding in e.g. FIRE, where Nopos is also reported in Figure 1: while performance degrades, it slowly becomes worse, moreover Nopos outperforms in generalization RoPE which is not the case in your Table above. Furthermore, in the paper "The impact of positional encoding on length generalization in transformers." it was shown that while Nopos degrades it outperforms others in the generalization, which is entirely opposite in your observations.
> > - I still don't understand how CAPE could generalize to unseen positions as $B$ which is input to $f$ is not seen during training, thus again we are in the regime of extrapolation / generalization, and not clear why function $f$ will be able to do better generalization in general.
> >
> > Having all above inconsistency with prior works reported results, this questions the validity of all results. Please, clarify your empirical findings which diverge from prior works.
> >
> > Thanks,
> >
> > AC

---

> ### Author Response · Authors · 2024-08-14
> **Response to Area Chair a2or (Part 1/2)**
>
> Dear Area Chair a2or,
>
> Thank you very much for your reply. We will answer your question below.
>
> For the question related to experiments:
> * **Our Table 2 CHE results (with 200K training steps) are almost aligned with the baseline[2].**
> * Previous work (such as Alibi) uses the sliding-window evaluation protocol, which divides sentences with L tokens and then evaluates. This will make the performance look better.
> * Previous works (such as Kerple) also use non-overlapping evaluation, while we choose the last 256 tokens so that the evaluation is more critical. Our Kerple implementation comes from its released code.
>    * **Kerple's paper uses non-overlapping evaluation: To train on or evaluate a sequence longer than $L$ tokens (which is usually the training length), it is typical
> to segment the sequence into L-length subsequences and train on or evaluate them independently. This will make the performance look better, compared to our evaluation protocol. For details, please refer to Alibi paper**
> * **With the different evaluation protocols and training settings, the reported results may be different, such as the Kerple performance difference between Kerple paper and FIRE paper**. In Kerple's paper, its ppl gradually decrease with the evaluation length increases. However, the FIRE paper Figure 6 (train length 512) reports that the Kerple method ppl gradually increases with the evaluation length increases.
> * **Moreover, even if we directly use previous works' results, our proposed method (with a more critical evaluation protocol so that the baseline is lower) could still achieve better performance.**
>    * For example, on the arxiv dataset Kerple paper reports that it could achieve 5.01 ppl on evaluation length 8192 (with training length 512), and our proposed method achieves 3.70 ppl on evaluation length 8192 (with training length 512).
>    * This suggests that even with a more critical evaluation protocol, and proposed method could still achieve good performance.
> * All our experiments are fair and consistent. **We directly use the popular GPT-NeoX framework for our experiments (it has Alibi implementation), and we promise that we will release the code. We are sure that the experiment setting is fair and the experiment results are reproducible.**
>
> **Q1: Why then in your comparison in Figures 2, 3, 4 and above Tables FIRE is the worst in the generalization? ? Even worse than RoPE?**
>
> A1: **This may be caused by the different training strategies and settings**.
>
> For our experiments, we train with 50K with length 512 (following Kerple setting) and 8 GPUs, while FIRE runs 600K steps with length 2048 and 128 GPUs. The FIRE uses MLP to predict the position encoding so it may need more training tokens and other potential training strategies.
>
> Also, **FIRE performance is actually better than RoPE, with a larger training length**
> |   Books Dataset (Train Length 512)         | 512   | 1024  | 2048  | 4096  | 8192  |
> |------------|-------|-------|-------|-------|-------|
> | RoPE|19.62|170.93|336.76|426.50|559.13|
> | FIRE     | 19.67|19.51|34.95| 129.33|278.783|
>
> **Our FIRE implementation**: we directly utilize the FIRE implementation code provided in FIRE's paper.
>
> **Q2: Why did you decide to select 128 context lengths for training?**
>
> A2: We do present the results of training length 512 (Figire 2) and 1024 (Figure 3) in our paper, which follows the setting of Alibi paper and Kerple paper.
>
> **Why choose 128?**
> The reason is that we focus on length extrapolation so we would like to know based on the training length $T_{train}$, how long our proposed method could still achieve good performance on evaluation length $T_{eval}$. Basically, there are two ways:
> * Train on the longer length and then evaluate on the longer length. However, this will take a lot of computing cost, as we know that the transformer cost is $N^2$, while the N is the sequence length.
> * The second way is that we train on a relatively short length (such as 128) and evaluate on a relatively longer length (such as 8192). This could reduce the computational cost.
> * Based on our hardware, we additional train on 128.
>
> **Q3: Could you explain the choice of Arxiv and Books3 data as e.g. prior works used WikiText, C4, OpenWebText2, GitHub?**
>
> A3: The Arxiv dataset and Books dataset are also commonly used in previous works, and **their Mean Document Size is relatively bigger.**
> * The mean document size: Wikipedia (en) (**1.11 KiB**), OpenWebText2 (**3.85 KiB**), GitHub (**5.25 KiB**)
> * The Arxiv dataset is commonly used in previous works, such as Kerple, FIRE and BiPE. The Mean Document Size of Arxiv is **46.61 KiB.**
> * The Books dataset is also commonly used in previous works, such as LongRoPE. Also, The Mean Document Size of Books is **538.36 KiB**, which is almost the largest one in Pile dataset.
> * As we focus on length extrapolation, we prefer to validate our proposed method on the long-context situations so that Arxiv and Books3 are preferred.

---

> ### Author Response · Authors · 2024-08-14
> **Response to Area Chair a2or (Part 2/2)**
>
> **Q4: The performance of Kerple**
>
> A4: This may be caused by the evaluation protocol or training strategy.
>
> For our evaluation protocol, we select the last 256 tokens in the sequence to evaluation so that the evaluation is more critical and could better reflect the model performance with the current sequence length.
>
> **According to the Kerple paper, you may find that its reported results achieves relatively better performance than our paper, FIRE and BiPE paper[1]**. For example, the RoPE result in Kerple's paper achieves 5.94 ppl with an evaluation length of 2048 on the Github dataset (training length is 512), while BiPE reports that RoPE achieves 15 ppl with an evaluation length of 2048 on the Github dataset (training length is 1024). This may be caused by different evaluation protocols or training strategies.
>
> * **Moreover, even if we directly use previous works' results, our proposed method (with a more critical evaluation protocol and the baseline is lower) could still achieve better performance.**
>    * For example, on the arxiv dataset Kerple's paper reports that it could achieve 5.01 ppl on evaluation length 8192 (with training length 512), and our proposed method achieves 3.70 ppl on evaluation length 8192 (with training length 512).
>    * This suggests that even with a more critical evaluation protocol, and proposed method could still achieve good performance.
>
> **Q5: how all results on 128 length are helpful for overall conclusions and claims of the paper**
>
> A5: **It suggests that our proposed method has the potential to extrapolate 64 times, compared to the training length**
>
> We would like to know the context boundary. Because of the limitation of computational resources, we would like to know the largest many times that our proposed method could extrapolate. With a training length of 128, we successfully extrapolate to 8192, suggesting the potential that our proposed method could extrapolate 64 times the training length.
>
> **Q6: The question for prior papers and Figure 5**
>
> A6: We will answer the question in two parts: 1) The question for the prior paper; 2) The Figure 5 baseline is Kerple, and thank you very much for your notice.
>
> **The question for the prior paper**
> As we mentioned above, the prior paper may use different evaluation protocols and experiment settings so that their performance may be better. For example, the Alibi uses the sliding-window evaluation protocol.
>
> For the previous paper that claims they could extrapolate to 16K. **We may pay attention to the FIRE paper. According to FIRE figure 6, we can find that it is hard for the prior paper method to keep performance on evaluation length 8192 (when the training length is 512).**
>
> **For Figure 5 caption**
>
> Thank you very much for your notice. The Figure 5 caption should be CAPE-Kerple but not CAPE-Alibi. We will revise it in the final revision.
>
> **Q7: Question about NoPE**
>
> A7: The training setting may be different. As we said above, the NoPE may need more training tokens and length. For our experiments, we train with 50K with length 512 (following Kerple setting) and 8 GPUs on the Arxiv or Books dataset, while FIRE runs 600K steps with length 2048 and 128 GPUs.
>
> Also, as we said above, the NoPE implicitly learns the position information so that we may sufficiently train the model so that the NoPE can present its ability to outperform other methods.
>
> **Q8: Not clear why the function will be able to do better generalization in general.**
>
> A8: We further conduct an ablation study on the $f$, proving that **$f$ help enhance the bias matrix.** CAPE improves the bias matrix for $A_{final}$, while the $A_{final}$ is used to calculate $A_{final}K$. **For the unseen position, the $B$ partially could handle it (FIRE changes the problem to interpolation), but is not accurate enough so that CAPE helps enhance the bias matrix $B$ via attention score**
> |   Books Dataset (Train Length 512)         | 512   | 1024  | 2048  | 4096  | 8192  |
> |------------|-------|-------|-------|-------|-------|
> | $QK^T+B$  (baseline) |19.68|19.06|20.44| 28.34|39.31|
> | $QK^T+B+f(B)$     | 19.64  | 18.83  | 18.49  | 20.62  | 23.49   |
> | $QK^T+B+f(QK^T, B)$     | 19.22| 18.22| 17.15| 17.63| 17.88|
> The experiment suggests two points:
> * The CAPE $f(QK^T, B)$ is better than naive $f(B)$, suggesting that the context-adaptive is important.
> * The $QK^T+B+f(B)$  is better than $QK^T+B$, suggesting that benefiting from improving the expressiveness of the bias matrix.
>
> **We promise that we will release the code. We are sure that the experiment setting is fair and the experiment results are reproducible**
>
> **Thank you very much for your constructive comments. If there is any further question, please let us know.**
>
> Reference
>
> [1] He, Z., Feng, , ... & He, D. Two Stones Hit One Bird: Bilevel Positional Encoding for Better Length Extrapolation. ICML, 2024.
>
> [2] Ruoss, A., , ... & Veness, J. (2023). Randomized positional encodings boost length generalization of transformers.

---

### Author Response · Authors · 2024-08-14
**Summarizing the Contributions of this Paper**

Dear ACs and Reviewers,

Thank you all for your time and effort in reviewing this paper. We are grateful for the ACs to arrange the discussion. We are grateful for the positive recognition by CfZF, NzBk and QhvW. Our contributions are well recognized by reviewers CfZF, NzBk and QhvW. Our experiment results are endorsed by reviewers CfZF, NzBk and QhvW, and our paper presentation is favored by reviewers tTna, CfZF, NzBk and QhvW.

Additionally, we outline our paper's main contributions, including the additional conclusions during the rebuttal discussion phase:
* We proposed CAPE, a novel approach to length extrapolation, which utilizes the attention score to dynamically adjust the position encoding.
* We conduct extensive experiments to validate the proposed method CAPE. We conduct experiments with different training lengths (128, 512, and 1024), different datasets(Arxiv, Books, and CHE benchmark), and different model sizes (125M, 350M, 2.7B, and 6.7B) and different evaluation metrics (perplexity metrics and accuracy metrics)
* Our proposed method is successfully trained on length 128 or 512, and then keeps good performance on evaluation length 8192.
* We promise that we will incorporate our discussion part into the paper's final revision, including but not limited to different model size experiments, the analysis of the computation cost, the visualization of CAPE, the discussion of context-boundary, the discussion of RoPE, and so on.

**The CAPE implementation is shown in Appendix G. Within less than 10 minutes of modification, CAPE (such as CAPE-Kerple or CAPE-FIRE) can be easily and directly used for your own transformer experiments (CAPE could improve performance for both within training length and beyond training length). We will release all the code for reproducing the experiment results. We promise that our experiment settings are fair and consistent.**

Again, we thank the ACs and all the reviewers for actively engaging in the rebuttal discussion and for their positive recognition of our work.

---

### Public Comment · ~Mingyu_Xu1 · 2024-12-06
**About the computation cost**

Hello, authors！

I am very interested in your paper.  The experiments seem to be conducted in relatively short context length. But now many LLM are pre-trained on at least 8k length. And many open-source models are already in a context of 128k. If DAPE can be efficiently implemented in a large context length, I believe this will enable DAPE to have a wider range of applications.

Can you provide further explanation on the computational complexity of DAPE? e.g., the comparison of computation of different position embedding by using Llama3-8b with a context of 128k.

Best regards,
Mingyu

---

> ### Public Comment · ~Chuanyang_Zheng3 · 2024-12-06
> **Response to Mingyu**
>
> Dear Mingyu Xu,
>
> Thank you very much for your interest in our work. We will answer your question below.
>
> **The cost could be divided into two parts: 1) computation cost: 2) real time cost**
>
> **The Computation Cost**
>
> * The cost of Feed-Forward Network is: $O(Nd_{head}^2d_{hidden}^2)$=$aNd_{head}^2d_{hidden}^2$, where a is a constant, N is the sequence length, $d_{head}$ is the attention head number and $d_{hidden}$ is the dimension for attention calculation.
> * The cost of Attention: $O(N^2d_{head}d_{hidden})$=$bN^2d_{head}d_{hidden}$, where b is a constant.
> * **The additional cost of DAPE**: $O(N^2d_{head}d_{DAPE})$=$cN^2d_{head}d_{DAPE}$, where c is a constant.
>
> **The Real Time Cost: Bontecck is I/O(Read In/ Read Out) when the length is very large, if without flash attention.**
>
> If without flash attention, we have to read the matrix with size $T \cdot T$ for more several times. Hence, the time cost bottleneck becomes the I/O time cost.
>
> Moreover, our poster will be presented next week at the NeurIPS conference. We could have a face-to-face discussion, if necessary.
> * **Poster Location:** East Exhibit Hall A-C #3009
> * **Poster Time:** Wed 11 Dec 11 a.m. PST — 2 p.m. PST
>
> Again, thank you very much for your attention, and hope to see you soon.
>
> Best regards,
>
> Chuanyang

---

### Public Comment · ~Chuanyang_Zheng3 · 2024-12-06

Dear Area Chair, Reviewers and any Researchers interested in this work,

Thank you very much for your attention and support to this work. If you have any questions, please let us know.

And we will also present the work next week so that we can have a face-to-face discussion, if necessary. The following are the details of the poster.

* **Poster Location: East Exhibit Hall A-C #3009**
* **Poster Time: Wed 11 Dec 11 a.m. PST — 2 p.m. PST**

Again, thank you very much for your attention, and hope to see you soon.

Best regards,

Chuanyang

---

### Decision · Program_Chairs · 2024-09-25

**Decision:**

Accept (poster)

**Comment:**

**Summary**

Authors propose a new positional embedding which consists of an additional mlp layer learnt on top of existing additive bias (expresses the relative positions) and the attention itself. Authors show that proposed context-aware positional embedding improves generalization over the baselines when applied on top of AliBi, KERPLE and FIRE for the sequences which are longer than training ones. Also authors show that proposed modification is learning both local and anti-local information.

**Justification for decision making**

It was a long and productive discussion among reviewers and authors, and also between AC (me) and authors. Most of the issues, concerns and ablations were resolved during long discussion and e.g. one of the reviewers raised the score from 6 to 8. However, 1/4 reviewers did not comment anything to rebuttal and has the lowest score of 3: I believe authors provided proper and reasonable answers after which the argumentation for rejection based on that review is no longer valid to me.

I raised several important issues and misalignments for the paper during the discussion period. During rebuttal period authors provided explanations to my points saying that they changed the evaluation protocol compared to prior works and argued that this protocol is more suited. They pointed out that this change in the protocol led to different results for the baselines compared to prior works results for that baselines. During AC-reviewers discussion one of the reviewers agreed to the raised issues, but also confirmed that the protocol authors used is more suitable. However the reviewer also agreed that better apple-to-apple comparison would strengthen the paper. In the end this reviewer is still on the positive side of the paper acceptance and experienced similar observations on many aspects as authors reported.

**Recommendation**

Based on the positive narrative of 3/4 reviewers (6, 7, 8), taking into account rebuttal to the 4th reviewer which did not respond to the rebuttal, also taking into account AC-authors discussion and AC-reviewers discussion, I **recommend acceptance of the paper**.

However, I ask authors for **several critical modifications for the camera-ready paper**:
- **change the title of the paper** due to collision with already existing work on positional embedding called CAPE and published at NeurIPS: Likhomanenko, T., Xu, Q., Synnaeve, G., Collobert, R. and Rogozhnikov, A., 2021. CAPE: Encoding relative positions with continuous augmented positional embeddings. Advances in Neural Information Processing Systems, 34, pp.16079-16092. to avoid naming problems in the future.
- include detailed discussion of evaluation protocols and why results for the baselines are different from the prior works to avoid misinterpretation of results and doubting the baselines from the readers, also for the sake of science.
- include all rebuttal discussions and ablations with new results into appendix as they all have merits and contribute to better understanding and practical applications of the proposed method.
- if possible add at least one apple-to-apple comparison and setup with prior works so as to have strong support that the baselines implementation is correct and consistent with prior works. I and reviewers still found some results reported for the baselines to be strange even when 512 training length is used (128 length definitely gives different observations compared to 512 training length and thus different from prior works as they all used larger context length for training).
- if the numbers and behaviour you observe diverges from prior work, please discuss this in the results and point to the source of this discrepancy.